# Development strategy of early childhood music education industry: An IFS-AHP-SWOT analysis based on dynamic social network

**Yuanyang Yue, Xiaoyan Shen** [ID]*

School of Early Childhood Education, Shanghai Normal University Tianhua College, Shanghai, China

* thth431@163.com

**Data Availability Statement:** All relevant minimal data are within the manuscript and its Supporting information files.

## Abstract

Early childhood music education has garnered recognition for its unique contribution to cognitive, emotional, and social development in children. Nevertheless, the industry grapples with numerous challenges, including a struggle to adapt traditional educational paradigms to new curriculum reforms, and an excessive emphasis on skill training at the expense of nurturing a love for music and aesthetics in children. To navigate these challenges and explore growth strategies for the early childhood music education industry, we initiated a comprehensive approach that involved distributing surveys to practitioners and parents and engaging experts for insightful discussions. Consequently, we proposed an analytical method based on dynamic social networks in conjunction with Intuitionistic Fuzzy Sets (IFS), Analytic Hierarchy Process (AHP), and Strengths, Weaknesses, Opportunities, and Threats (SWOT) analysis, collectively referred to as IFS-AHP-SWOT. This integrated methodology synergizes the capabilities of dynamic social networks, IFS, AHP, and SWOT analysis to offer a nuanced perspective on industry development strategies. The findings underscore that institutions within the early childhood music education industry need to adopt a development strategy that leverages their strengths and opportunities to foster sustainable growth. Ultimately, this research aims to provide critical decision-making support for industry practitioners, policymakers, and researchers, contributing significantly to the ongoing discourse on strategic development in the early childhood music education industry.

## Section 1: Introduction

In recent years, early childhood music education has been increasingly recognized for its significant role in promoting cognitive, emotional, and social development in young children [1, 2]. For example, in October 2020, China issued the "Opinions on Strengthening and Improving School Aesthetic Education in the New Era", emphasizing that beauty is an important source of moral purity and spiritual richness. School aesthetic education curriculum is centered around art, including music, fine arts, calligraphy, dance, drama, opera, film and television courses. In pre-school education, art activities suitable for the physical and mental characteristics of young children are carried out, focusing on cultivating children's beautiful

**Funding:** The author(s) received no specific funding for this work.

**Competing interests:** The authors have declared that no competing interests exist.

and kind hearts and the ability to cherish beautiful things. For young children, a diverse aesthetic education can better observe and feel the world, resulting in genuine emotional expression and personal experience, which is crucial for individual learning and development [3]. In other words, music education plays a key mediating role in the entire education and teaching process, which is something other teaching methods and means cannot achieve [4, 5]. Therefore, to promote the all-round development of young children, it is imperative to carry out music education.

While there is a relatively large amount of research and findings on early childhood music education currently, there are still some issues. Firstly, in basic education activities, traditional education is facing unprecedented challenges due to the impact of the new curriculum reform, and there is an urgent need to innovate educational concepts [6]. Secondly, since the reform of basic education, all kindergartens, including private and public, are facing new educational reform challenges [7]. Nevertheless, in teaching practice, there is often an excessive emphasis on skill training, disregarding the nurturing of children's interest in music and their aesthetic abilities. During music activities, the focus typically remains on teaching children to sing and play games, overlooking the exploration of other facets of music. The single teaching model makes it impossible for teachers to fully understand children's musical performance abilities, which makes it impossible to further expand musical activities [8]. For this reason, kindergarten music teaching and education cannot meet the needs of children's overall development, and it is difficult to fully tap into the potential of each child. The ultimate goal of music education is not only to train singers or pianists, and it's not only focused on the training of techniques and skills. The key is to use music to enhance children's aesthetic ability, stimulate their interest in learning music, and cultivate them at a spiritual level, thereby continuously stimulating individual creativity and imagination. At the same time, music education plays an irreplaceable important role in the physical and mental development of children. Through music activities, it can help enhance children's self-cognition, interpersonal communication, physical coordination, and language expression capabilities [9–11].

With the rapid development of the early childhood music education industry, conducting effective strategic analysis to promote its sustainable high-quality growth has become particularly important. SWOT analysis is a commonly used method for business and industry strategic analysis [12, 13]. However, because it cannot determine the degree of influence of each factor, it is often used in combination with AHP (AHP-SWOT method) [14–16]. Although AHP has proven to be effective and simple in dealing with multi-criteria decision-making problems, it cannot fully address inherent uncertainties and fuzziness. Therefore, this paper introduces Intuitionistic Fuzzy Set Theory into the AHP-SWOT method (IFS-AHP-SWOT method) to compensate for the limitations caused by experts' limited knowledge and subjective evaluation criteria [17, 18]. Strategic analysis for the early childhood music education industry requires the collaborative judgment of multiple experts. Therefore, this paper improves the method of aggregating the judgment information of multiple experts while considering social network relationships formed by different trust relationships, addressing the potential abnormal results due to individual differences [19, 20]. This improved IFS-AHP-S-WOT analysis method based on dynamic social networks, combined with the strategic analysis viewpoints of educators, practitioners, and policymakers, provides strong support for the sustainable development of the early childhood music education industry. The research results indicate that institutions in the early childhood music education industry should adopt development strategies based on strengths and opportunities (SO). This study comprehensively applies dynamic social networks, IFS, AHP, and SWOT analysis methods to offer a systematic analytical framework and guiding recommendations for the development strategy of the early childhood music education industry. It is hoped that this research will provide valuable

reference for early childhood music education practitioners, policymakers, and researchers, promoting the continuous development and progress of the industry.

The rest of the paper is organized as follows: "Section 2: Literature Review" reviews the relevant literature. "Section 3: Model" defines the related model method. "Section 4: SWOT Factor Identification" shows the SWOT analysis factors given by experts. "Section 5: Numerical analysis process" presents the numerical calculation process. In "Section 6: Conclusions", this paper compares the equilibrium results under different conditions and draws conclusions.

## Section 2: Literature review

The literature relevant to this paper focuses on two aspects: one is early childhood music education, and the other is research methods such as dynamic social networks, intuitionistic fuzzy evaluation, AHP, etc.

In the field of early childhood music education, the study by [21] found that musical skills are closely related to phonemic awareness and reading development, suggesting that musical perception utilizes auditory mechanisms associated with reading. Niland [22] found that child-centered musical games are an effective way to help young children explore musical elements and concepts within the context of play. Harper [23] suggested that explicit teaching of Western nursery rhymes can enhance children's phonemic awareness. Rauscher and Hinton [24] found that music instruction can enhance children's spatial-temporal reasoning, numerical reasoning, and phonemic awareness, with early-trained musicians benefiting the most. Research by François et al. [25] showed that music training can directly promote phoneme segmentation, thus playing a significant role in child language development. Chobert et al. [26] confirmed the positive impact of music training on children's language skills, highlighting the importance of music training in child education and providing new remediation strategies for children with language learning difficulties. Sallat and Jentschke's [27] research found that children with Specific Language Impairment (SLI) performed worse in music perception, highlighting the close relationship between language acquisition and music processing. According to Linnavalli et al. [28], continuous participation in group music activities can have a positive impact on the language skills of preschool children. Zuk et al. [29] found that music training is associated with enhanced activation in the bilateral temporo-parietal regions, which may aid reading development. Bowmer et al. [30] found that music training has a positive effect on executive functions in young children, but there is still debate on issues such as intervention duration, experimental design, and target age group. Research by Fasano et al. [31] showed that short-term orchestral training can promote the development of children's inhibitory control, reduce self-reported hyperactivity levels, which is significant for music education and education. Loui et al. [32] found that music training is associated with children's IQ, language abilities, and the microstructural properties of white matter in the brain, suggesting that music practice may improve cognition and brain health. Pitt's [33] research encouraged children with communication difficulties to participate in interactive games through interdisciplinary methods and found that musical activities help to improve children's communication abilities. Welch's [34] research explored the potential and actual benefits of musical activities for young children but also pointed out that many teachers might lack professional knowledge in organizing effective music education. Thapa and Rodriguez-Quiles's [35] study emphasized the significant role of music education in child development and proposed investing in it as a core subject.

In terms of research methods, for social network decision-making studies, Wu et al. [19] proposed a group decision-making model based on social networks that determine weights by considering trust and knowledge levels to optimize decision-making and find the best solution.

Wu et al. [20] proposed a new social network group decision-making framework that improves the consistency of group decision-making at the least cost by optimizing the trust relationship model and feedback mechanism. Dong et al. [36] reviewed the process of consensus building in social network group decision-making, classified it into two paradigms based on trust relationships and opinion evolution, and pointed out the challenges for future research. Liu et al. [37] introduced a new large-scale group decision-making model that reduces conflicts and optimizes decisions through steps such as trust propagation, conflict detection and elimination, and the determination of decision-maker weights, and the practicality of the model has been empirically validated. Gai et al. [38] proposed a joint feedback strategy framework that combines social network context and feedback behavior to improve the efficiency of large-scale group decision-makers reaching consensus, and through numerical and comparative analysis validated the effectiveness of this strategy under different feedback behaviors. Then, for intuitionistic fuzzy and AHP research, Sadiq and Tesfamariam [39] proposed an improved intuitionistic fuzzy analytic hierarchy process (IFAHP), which can handle the fuzziness and uncertainty in environmental decision-making processes, and proved the effectiveness of this method in selecting the optimal drilling fluid through examples. Xu and Liao [40] proposed a novel IFAHP that better handles preference uncertainty and introduced a method for automatically repairing inconsistent preference relations, which can improve efficiency without the involvement of decision-makers. Tavana et al. [17] proposed a method for evaluating and ranking reverse logistics suppliers using SWOT analysis and IFAHP and found that focusing on core business is more important than cost reduction. In the context of the global energy crisis, Abdullah and Najib [41] used the new IFAHP for sustainable energy planning and found that nuclear energy is the best energy solution. The above research can be summarized as shown in Table 1, as follows:

In summary, current research on early childhood music often focuses on the promotion of various aspects of music education for young children, such as social and perceptual benefits. However, it frequently overlooks the broader, long-term development strategy of the early childhood music education industry within the context of the current dynamic social environment. This gap in research leaves a critical need for a comprehensive approach to address the industry's future growth and adaptability.

One of the shortcomings in the current research landscape is the prevalent use of intuitive and somewhat disjointed methods, such as intuitionistic fuzzy AHP (Analytic Hierarchy Process) and SWOT (Strengths, Weaknesses, Opportunities, and Threats) analysis. These methods are often employed independently, failing to harness the full potential of their complementary attributes. Moreover, they generally disregard the significant impact of social networks and interactions among expert reviewers in the process of shaping industry development strategies.

To address these limitations, this paper proposes a novel approach that combines dynamic social networks, IFS (Intuitionistic Fuzzy Set), AHP, and SWOT analysis methods. This integrated methodology aims to establish a comprehensive and cohesive framework for analyzing the development strategy of the early childhood music education industry. By merging these

**Table 1. Summary of literature on research methods.**

|  | SWOT | AHP | Fuzzy decision-making | Social network |
|---|---|---|---|---|
| [19, 20, 36–38] |  |  | ✓ | ✓ |
| [39, 40] |  | ✓ | ✓ |  |
| [17, 41] | ✓ | ✓ | ✓ |  |
| This paper | ✓ | ✓ | ✓ | ✓ |

diverse techniques, this research endeavors to create a more holistic perspective on industry growth and provide a more effective and adaptable strategy. It recognizes the importance of considering both the intrinsic factors identified through SWOT analysis and the dynamic, interrelated social factors that shape the industry's trajectory, as assessed through social network analysis. In doing so, it offers valuable guidance for stakeholders and decision-makers within the early childhood music education sector to make informed, data-driven decisions that can lead to sustained success in a rapidly evolving environment.

## Section 3: Model

### 3.1 Intuitionistic Fuzzy Numbers (IFN) theory

Intuitionistic Fuzzy Numbers (IFN) is a mathematical approach that extends the concept of fuzzy numbers, allowing for a more accurate representation of uncertainty in the real world. IFNs are composed of two functions: membership function and non-membership function. Compared to traditional fuzzy numbers, IFNs are better suited to handle uncertainty and ambiguity in real-world scenarios. The relevant definitions and properties are as follows:

**Definition 1**. Let $X = \{x_1, x_2, \cdots, x_n\}$ be a fixed set. An intuitionistic fuzzy number $A$ in $X$ is defined as follows [42]:

$$A = \{\langle x, \mu_A(x), v_A(x)\rangle | x \in X\} \tag{1}$$

Where $\mu_A(x)$ and $v_A(x)$) represent the membership degree and non-membership degree of the element $x$, respectively, and $\pi_A(x) = 1 - \mu_A(x) - v_A(x)$ is referred to as the hesitancy of $x$ in the set $A$. These values satisfy the conditions: $\mu_A(x), v_A(x) \in [0,1]$ and $\mu_A(x) + v_A(x) \in [0,1]$ for all $x \in X$.

**Definition 2**. Let $a = (\mu_a, v_a)$ and $b = (\mu_b, v_b)$ be two IFNs. The distance between $a$ and $b$ is then defined as follows [43]:

$$dis(a, b) = \frac{(|\mu_a - \mu_b| + |v_a - v_b|)}{2} \tag{2}$$

Numerical examples of Formula 2 can be found in Appendix D (Appendix file in the S1 File): Example 1.

**Definition 3**. To rank IFNs, a function be proposed with the following mathematical form [44]:

$$\rho(a) = 0.5(1 + \pi_a)(1 - \mu_a) \tag{3}$$

The smaller the value of $\rho(a)$, the larger the IFN in terms of the quantity of positive information contained and the reliability of the information. Numerical examples of Formula 3 can be found in Appendix D (Appendix file in the S1 File): Example 2.

**Definition 4**. The operational rules between IFNs $a = (\mu_a, v_a)$ and $b = (\mu_b, v_b)$ are as follows:

$$a \oplus b = (\mu_a + \mu_b - \mu_a\mu_b, v_a v_b) \tag{4}$$

$$a \otimes b = (\mu_a\mu_b, v_a + v_b - v_a v_b) \tag{5}$$

$$\lambda a = \left(1 - (1 - \mu_a)^\lambda, (v_a)^\lambda\right), \lambda > 0 \tag{6}$$

$$a^\lambda = \left((\mu_a)^\lambda, 1 - (1 - v_a)^\lambda\right), \lambda > 0 \tag{7}$$

The numerical examples of Formulas 4 and 5 can be found in Appendix D (Appendix file in the S1 File): Example 3.

## 3.2 Intuitionistic fuzzy preference relations

Assuming there are $N$ alternative options (represented as the set $A = \{1, 2, \cdots, N\}$), $M$ experts (represented as the set $E = \{1, 2, \cdots, M\}$) evaluate the preference relations of these alternative options using intuitionistic fuzzy numbers, forming the intuitionistic fuzzy preference relation matrices for the alternative options. Expert $e$ perceives the intuitionistic fuzzy preference relation between option $i$ and option $j$ as $b_{ij}^e = \left(u_{ij}^e, v_{ij}^e\right)$, where $u_{ij}^e$ represents the degree to which option $i$ is preferred over option $j$, $v_{ij}^e$ represents the degree to which option $i$ is not preferred over option $j$, and $\pi_{ij}^e = 1 - u_{ij}^e - v_{ij}^e$ is interpreted as the degree of uncertainty or hesitation, satisfying the conditions: (1) $u_{ij}^e, v_{ij}^e \in [0, 1]$; (2) $u_{ij}^e + v_{ij}^e < 1$; (3) $u_{ij}^e = v_{ji}^e, u_{ji}^e = v_{ij}^e, u_{ii}^e = v_{ii}^e = 0.5$.

The intuitionistic preference relations provided by experts need to satisfy consistency. Lack of consistency in preference relations can lead to incorrect solutions. Multiplicative consistency is a crucial property of preference relations, and it is defined as follows:

An intuitionistic preference relation $B^e = \left[b_{ij}^e\right]_{N \times N}$ with $b_{ij}^e = \left(u_{ij}^e, v_{ij}^e\right), i, j = 1, 2, \cdots, N$ is multiplicative consistent if

$$u_{ij}^e = \begin{cases} 0, if \left(u_{ik}^e, u_{kj}^e\right) \in \{(0, 1), (1, 0)\} \\ \dfrac{u_{ik}^e u_{kj}^e}{u_{ik}^e u_{kj}^e + \left(1 - u_{ik}^e\right)\left(1 - u_{kj}^e\right)}, otherwise\ for\ all\ i \leq t \leq k \end{cases} \tag{8}$$

$$v_{ij}^e = \begin{cases} 0, if \left(v_{ik}^e, v_{kj}^e\right) \in \{(0, 1), (1, 0)\} \\ \dfrac{v_{ik}^e v_{kj}^e}{v_{ik}^e v_{kj}^e + \left(1 - v_{ik}^e\right)\left(1 - v_{kj}^e\right)}, otherwise\ for\ all\ i \leq t \leq k \end{cases} \tag{9}$$

We can return the inconsistent preference relations to experts for re-evaluation, but this approach poses some challenges. Experts may be unable to provide consistent preference relations due to limitations in their knowledge domain, lack of interest, or time constraints. Therefore, we need to rely on the following automated algorithms to address this issue.

**Algorithm 1**. The given intuitionistic fuzzy preference relation $B^e = \left[b_{ij}^e\right]_{N \times N}$ with $b_{ij}^e = \left(u_{ij}^e, v_{ij}^e\right), i, j = 1, 2, \cdots, N$, after undergoing algorithmic updates, satisfies multiplicative

consistency and is represented as $\bar{B}^e = \left[\bar{b}_{ij}^e\right]_{N\times N}$. When $j > i+1$, $\bar{b}_{ij}^e = \left(\bar{u}_{ij}^e, \bar{v}_{ij}^e\right)$, where

$$\bar{u}_{ij}^e = \frac{\left(\prod_{k=i+1}^{j-1} u_{ik}^e u_{kj}^e\right)^{\frac{1}{j-i-1}}}{\left(\prod_{k=i+1}^{j-1} u_{ik}^e u_{kj}^e\right)^{\frac{1}{j-i-1}} + \left(\prod_{k=i+1}^{j-1} \left(1-u_{ik}^e\right)\left(1-u_{kj}^e\right)\right)^{\frac{1}{j-i-1}}} \tag{10}$$

$$\bar{v}_{ij}^e = \frac{\left(\prod_{k=i+1}^{j-1} v_{ik}^e v_{kj}^e\right)^{\frac{1}{j-i-1}}}{\left(\prod_{k=i+1}^{j-1} v_{ik}^e v_{kj}^e\right)^{\frac{1}{j-i-1}} + \left(\prod_{k=i+1}^{j-1} \left(1-v_{ik}^e\right)\left(1-v_{kj}^e\right)\right)^{\frac{1}{j-i-1}}} \tag{11}$$

When $j = i+1$, $\bar{b}_{ij}^e = b_{ij}^e = \left(\bar{u}_{ij}^e, \bar{v}_{ij}^e\right)$, where $u_{ij}^e = \bar{u}_{ij}^e$ and $v_{ij}^e = \bar{v}_{ij}^e$

When $j < i$, $\bar{b}_{ij}^e = \left(\bar{v}_{ji}^e, \bar{u}_{ji}^e\right)$.

Although the above algorithm can transform an intuitionistic fuzzy preference matrix into a preference matrix with multiplicative consistency, if the difference between two preference matrices is significant, it indicates that the transformed preference matrix cannot convey the original decision awareness of the experts. Therefore, this study defines a measure to evaluate whether the preference matrix, obtained after algorithmic transformation and possessing multiplicative consistency, can convey the original decision awareness of the experts.

**Definition 8**: Given $B^e$ as the intuitionistic preference relation, $\bar{B}^e$ is referred to as the multiplicative consistent intuitionistic preference relation if it satisfies the following conditions. In that case, $\bar{B}^e$ can be considered an acceptable multiplicative consistent intuitionistic preference relation [44].

$$D(B^e, \bar{B}^e) = \frac{1}{2(N-1)(N-2)} \sum_{i=1}^{N}\sum_{k=1}^{N}\left(\left|\bar{u}_{ij}^e - u_{ij}^e\right| + \left|\bar{v}_{ij}^e - v_{ij}^e\right| + \left|\bar{\pi}_{ij}^e - \pi_{ij}^e\right|\right) < \varepsilon \tag{12}$$

Where $\varepsilon$ represents the threshold for multiplicative consistency.

Furthermore, although the intuitionistic fuzzy preference relation modified by Algorithm 1 can satisfy multiplicative consistency, it may not necessarily fulfill the acceptable multiplicative consistency. Therefore, to preserve the original decision preferences of the experts as much as possible while meeting the criteria for acceptable multiplicative consistency, this study employs Algorithm 2 for further refinement.

**Algorithm 2**. Given an intuitionistic fuzzy preference relation $B^e$ and the intuitionistic fuzzy preference relation $\bar{B}^e$ modified by Algorithm 1, if $B^e$ and $\bar{B}^e$ fail to satisfy the conditions for acceptable multiplicative consistency (as defined in Eq 12), the intuitionistic fuzzy preference relation B is recombined with the intuitionistic fuzzy preference relation $B^e$, yielding [40]:

$$B^e := \varphi\left(B^e, \bar{B}^e\right) \tag{13}$$

$$u_{ij}^e := \frac{\left(u_{ij}^e\right)^{1-\theta}\left(\bar{u}_{ij}^e\right)^{\theta}}{\left(u_{ij}^e\right)^{1-\theta}\left(\bar{u}_{ij}^e\right)^{\theta} + \left(1 - u_{ij}^e\right)^{1-\theta}\left(1 - \bar{u}_{ij}^e\right)^{\theta}} \tag{14}$$

$$v_{ij}^e := \frac{\left(v_{ij}^e\right)^{1-\theta}\left(\bar{v}_{ij}^e\right)^{\theta}}{\left(v_{ij}^e\right)^{1-\theta}\left(\bar{v}_{ij}^e\right)^{\theta} + \left(1 - v_{ij}^e\right)^{1-\theta}\left(1 - \bar{v}_{ij}^e\right)^{\theta}} \tag{15}$$

Where $\theta$ is a control parameter determined by the decision maker. Through this algorithm, we are able to automatically enhance the consistency level of any intuitive preference relation while minimizing the loss of original information. This process is convergent, and the resulting intuitive preference relation exhibits weak transitivity.

### 3.3 Alternative weights and expert weights

**3.3.1 Alternative weights.** After expert $e \in E$ provides a preference matrix of $N$ alternative solutions ($B^e = \left[b_{ij}^e\right]_{N \times N}$ with $b_{ij}^e = \left(u_{ij}^e, v_{ij}^e\right), i, j = 1, 2, \cdots, N$), it is necessary to calculate the weights of each alternative. This is done to quantitatively assess the relative importance or desirability of each alternative in the decision-making process. The weight of alternative $i$, derived from the preference matrix of fuzzy intuitive preferences, is given by the following equation:

$$\omega_i = \left(\frac{\sum_{j=1}^N u_{ij}^e}{\sum_{i=1}^N \sum_{j=1}^N \left(1 - v_{ij}^e\right)}, 1 - \frac{\sum_{j=1}^N \left(1 - v_{ij}^e\right)}{\sum_{i=1}^N \sum_{j=1}^N u_{ij}^e}\right) \tag{16}$$

**3.3.2 Expert weights.** Social networks are relatively stable social systems formed by interactions among individuals. Social network analysis involves studying the social relationships among members of a group. Accurate quantitative analysis of these relationships can reveal the intrinsic connections between members and explain certain social phenomena. In a network of relationships, identifying the importance of a member involves analyzing the extent to which that member is central in the entire network. The degree of trust that a member $e_l$ has in another member $e_k$ is represented as $T_{lk} = \left(u_{lk}^t, v_{lk}^t\right)$, and the sum of member $e_l$'s relationships with all other experts is represented as $T_l$, defined as $e_l$'s IFN centrality:

$$T_l = \left(u_l^t, v_l^t\right) = \left(\frac{1}{|E/l|}\sum_{k \in E/l} u_{lk}^t, \frac{1}{|E/l|}\sum_{k \in E/l} v_{lk}^t\right) \tag{17}$$

The sum of all experts' IFN relationships is represented as:

$$T = \left(\frac{1}{|E|}\sum_{l \in E} u_l^t, \frac{1}{|E|}\sum_{l \in E} v_l^t\right) \tag{18}$$

The degree centrality of an expert is primarily used to measure their importance and influence in the entire network. A higher degree centrality value indicates more connections with other experts, signifying greater significance of the expert in the network [45]. During the Multi-Criteria Decision Making process, assigning weights to each expert reflects the importance of their respective decisions on the final outcomes. In the given group's trust relationship

network, the proximity coefficient between an expert $e_l \in E$ and the entire set of experts $E$ is defined as follows:

$$wt_l = 1 - dis(T_l, T) \tag{19}$$

The larger the value of $wt_l$, the closer the expert $e_l$ is to the center of the group, indicating higher importance. The weight of each expert can be defined as follows:

$$w_l = \frac{wt_l}{\sum_{l \in E} wt_l} \tag{20}$$

**3.3.3 Group consensus in dynamic social networks.** Group consensus refers to the collective agreement within a group. In order to integrate expert evaluation information for decision-making, a context-based IFS distance measure is employed to determine the closeness between members. This analysis helps assess the similarity between members and the group center, ensuring consistency among members [46, 47]. Assuming there are a total of $N$ candidate solutions, the comprehensive evaluation value of expert $e \in E$ for these $N$ candidate solutions is represented as:

$$a_i^e = \sum_{j \in A} \oplus \omega_j b_{ij}^e \tag{21}$$

The similarity between expert $e \in E$ and the group $E$ is represented as:

$$sim_{e,E} = \sum_{i=1}^{N} \left[ dis\left( a_i^e, \sum_{e \in E} w_e a_i^e \right) \right] \tag{22}$$

When the difference between expert e and the group E is significant, it may be necessary for expert e to modify their evaluation and inform other experts in the group. Other experts may also adjust their perception of expert e. If there are still members who have not reached the threshold after multiple modifications, their evaluation elements will be directly adjusted without considering their preferences. The specific expression is as follows:

$$H = \beta H_0 + (1 - \beta) G_0 \tag{23}$$

where $H_0$ and $G_0$ represent the opinions of the member before adjustment and the group decision information, respectively. $\beta$ is the adjustment coefficient. Generally, the larger the value of $\beta$, the more the member retains the original opinions, and the smaller the adjustment amplitude.

## 3.4 Decision-making steps

Step 1: Define the evaluation criteria system for SWOT analysis of the early childhood music education industry's development strategy. Then proceed to the next step.

Step 2: Have each strategic analysis expert use the IFS (Formula 1) to compare each criterion pairwise, creating multiple intuitionistic fuzzy preference relation matrices. Continue to the next step.

Step 3: Use Formula 12 to check the consistency of the experts' provided intuitionistic fuzzy preference relations. If all intuitionistic fuzzy preference relations are consistent and acceptable, move on to Step 5; otherwise, proceed to Step 4.

Step 4: Use Algorithm 1 and Algorithm 2 to resolve inconsistent intuitionistic fuzzy preference relations (or return the inconsistent relations to experts for reevaluation until they become acceptable). Then, proceed to the next step.

Step 5: Based on all the intuitionistic fuzzy preference relations, calculate the attribute weights for each criterion in the evaluation criteria system. Calculate expert weights based on social networks (Formulas 16–20) and move on to the next step.

Step 6: Utilize expert weights and attribute weights to aggregate all intuitionistic fuzzy preference relation matrices (Formulas 21–23) to obtain the group intuitionistic fuzzy preference relations. Check for group consensus; if achieved, continue to the next step. Otherwise, return to Step 2, where experts with significant differences in group intuitionistic fuzzy preference relations will reevaluate, while the remaining experts will reassess their social network relationships.

Step 7: Apply Formulas 3 to 7 to aggregate the group intuitionistic fuzzy preference relations and derive the final scores for S (Strengths), W (Weaknesses), O (Opportunities), and T (Threats).

The above decision-making steps can be summarized as shown in Fig 1.

## Section 4: SWOT factor identification

In order to analyze the development strategy of the early childhood music education industry, we used the Delphi method to construct a total of 13 SWOT factors based on the literature review, questionnaires (see the Appendix A, Appendix file in the S1 File, for details, a total of 500 questionnaires were distributed and 476 were returned) and the opinions of 7 industry-related experts (see Table 2 for relevant information of the experts).

Because the research design of this article is based on human interviews and surveys, the following statement is hereby made:

i. Our study begins with questionnaire distribution on May 15, 2022 and ends on January 30, 2023. Decision-making for 7 experts begins on February 17, 2023 and ends on March 14, 2023.

ii. All participants (7 experts) provided informed consent. What we obtained was written consent, and each participant signed an informed consent form before starting the study, which clarified the purpose, process, possible risks and benefits of the study, and their rights. Only a screenshot of the written consent is shown in the article (see Fig 10 in the Appendix A, Appendix file in the S1 File), and the scanned copy can be obtained from the author. Our study did not include minors. In addition, participants who received the questionnaire were free to choose whether to fill it out.

We believe that these factors fully represent the strengths, weaknesses, opportunities and threats of the current development of the early childhood music education industry (see Table 3).

### 4.1 Strengths

**S1. Psychological needs.** Music education has a positive impact on the emotional expression and psychological development of young children. According to psychological theories, music, as a medium of emotional expression, can help young children express their inner feelings, emotions, and affections, promoting self-recognition of emotions and emotional management skills [48]. Moreover, psychological research suggests that music education can enhance young

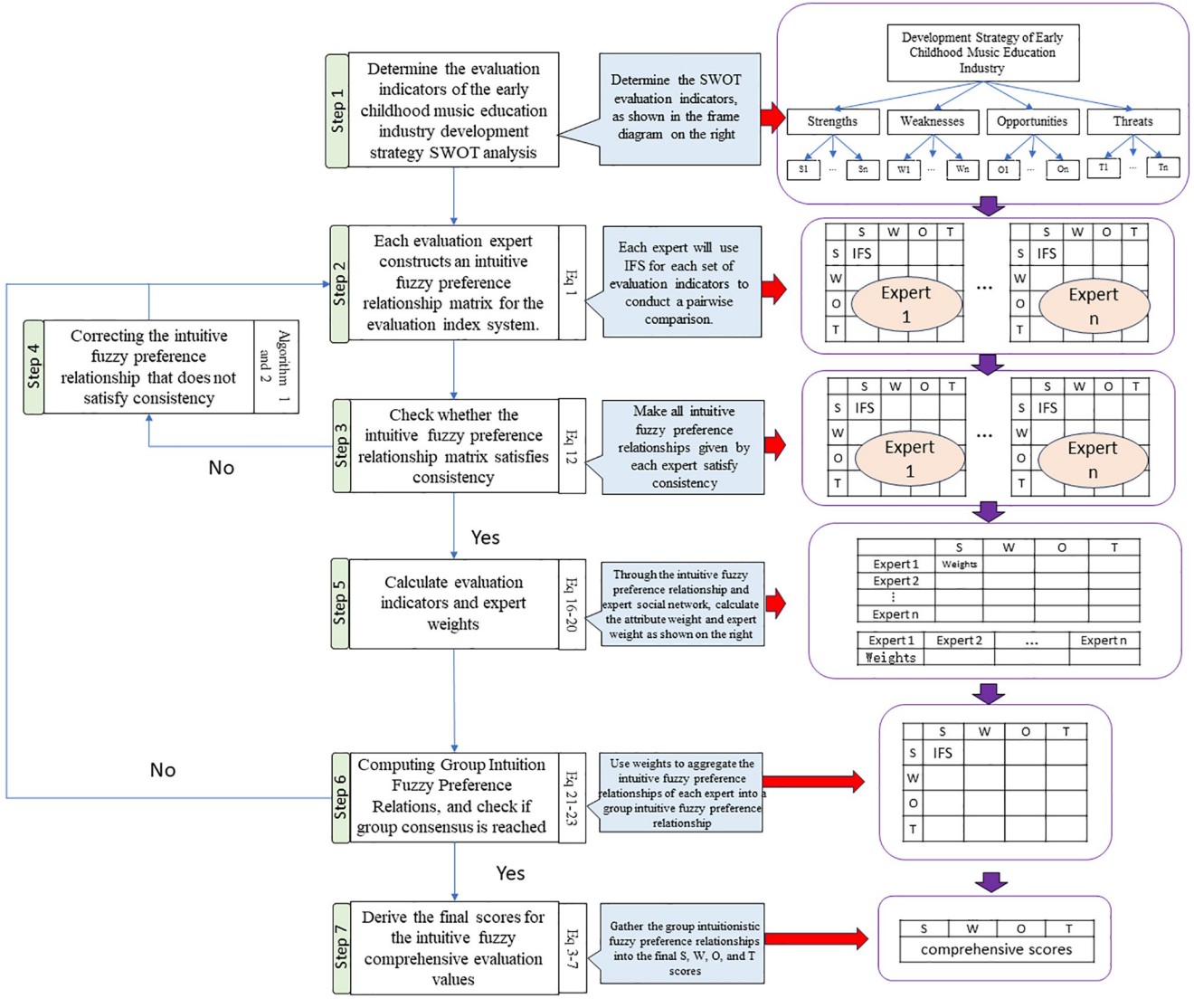

**Fig 1. Decision-making steps flow chart.**

**Table 2. Information about experts.**

| No. | Title | Affiliated institution | Professional field |
|---|---|---|---|
| 1 | Professor | Shandong Normal University School of Music Education | early childhood development |
| 2 | Associate Professor | Shandong University of Arts | Music Psychology |
| 3 | Music teacher | First Kindergarten of Shandong University | Early childhood music education |
| 4 | Music teacher | Primary School Affiliated to Shandong Normal University | Music Education and Cultural Studies |
| 5 | Music Psychologist | Jinan Xinyu Psychological Counseling Center | Music Psychology |
| 6 | Curriculum Designer | Jinan Yangxin Art | Curriculum Design |
| 7 | Marketing staff | Jinan Yiheng Art Training | Marketing |

**Table 3. Summary of identified SWOT factors.**

| Strengths | Weaknesses |
| --- | --- |
| S1. Psychological needs | W1. Industry competition |
| S2. Emphasis on art education | W2. Teacher shortage |
| S3. Market potential | W3. Brand influence |
| S4. Educational Innovation | W4. Regulatory restrictions |
| Opportunities | Threats |
| O1. Policy support | T1. Economic fluctuations |
| O2. Technological innovation | T2. Alternative products |
| O3. Market demand | T3. Regulatory risk |
| O4. Cross-border cooperation | T4. Industry Standards |
|  | T5. Intellectual Property |

children's emotional intelligence and social skills, improving their self-esteem and confidence [49]. Young children participating in music activities perform better in emotional expression and emotional management. Compared with other children, they are more likely to establish good emotional relationships with peers [50].

**S2. Emphasis on art education.** With society's increasing attention to art education, parents are putting more emphasis on cultivating their children's music literacy. According to educational theories, art education is crucial to the comprehensive development of young children, especially music education, which can promote the development of children's perception, creative thinking, and aesthetic ability [51]. In modern society, parents are increasingly focusing on cultivating their children's comprehensive qualities, including music literacy. According to the results of the questionnaire survey (Figs 2 and 3), the vast majority of parents expressed that they would invest significantly in early childhood music education, hoping to cultivate their children's interest in and understanding of music, and believe it is beneficial to the children's comprehensive development.

**S3. Market potential.** The early childhood education market is substantial, and music education, as part of it, has significant market potential. According to marketing theories, the early childhood education market is increasingly valued in modern society, and parents' demand for early childhood education is continuously increasing [52, 53]. Music education, as part of the field of early childhood education, meets parents' needs for the comprehensive development of their children. According to the results of the questionnaire survey (Figs 4 and 5), in recent years, the number of early childhood music education institutions has been increasing, with a considerable scale of students. Most parents believe that the market for early childhood music education has a broad development prospect.

**S4. Educational innovation.** Technological advances have created room for innovation in music education, such as online teaching and multimedia interaction. According to educational technology theories, the application of modern technology has brought new development opportunities to music education [54]. The emergence of online teaching platforms provides a broader learning pathway for early childhood music education, unrestricted by time and space, making music education resources more global and inclusive [55]. At the same time, the use of multimedia interactive technology makes music teaching more lively and interesting, increasing children's interest in music learning and participation [56].

## 4.2 Weaknesses

**W1. Industry competition.** The competition in the early childhood music education market has become particularly fierce, with the challenge of attracting and retaining customers

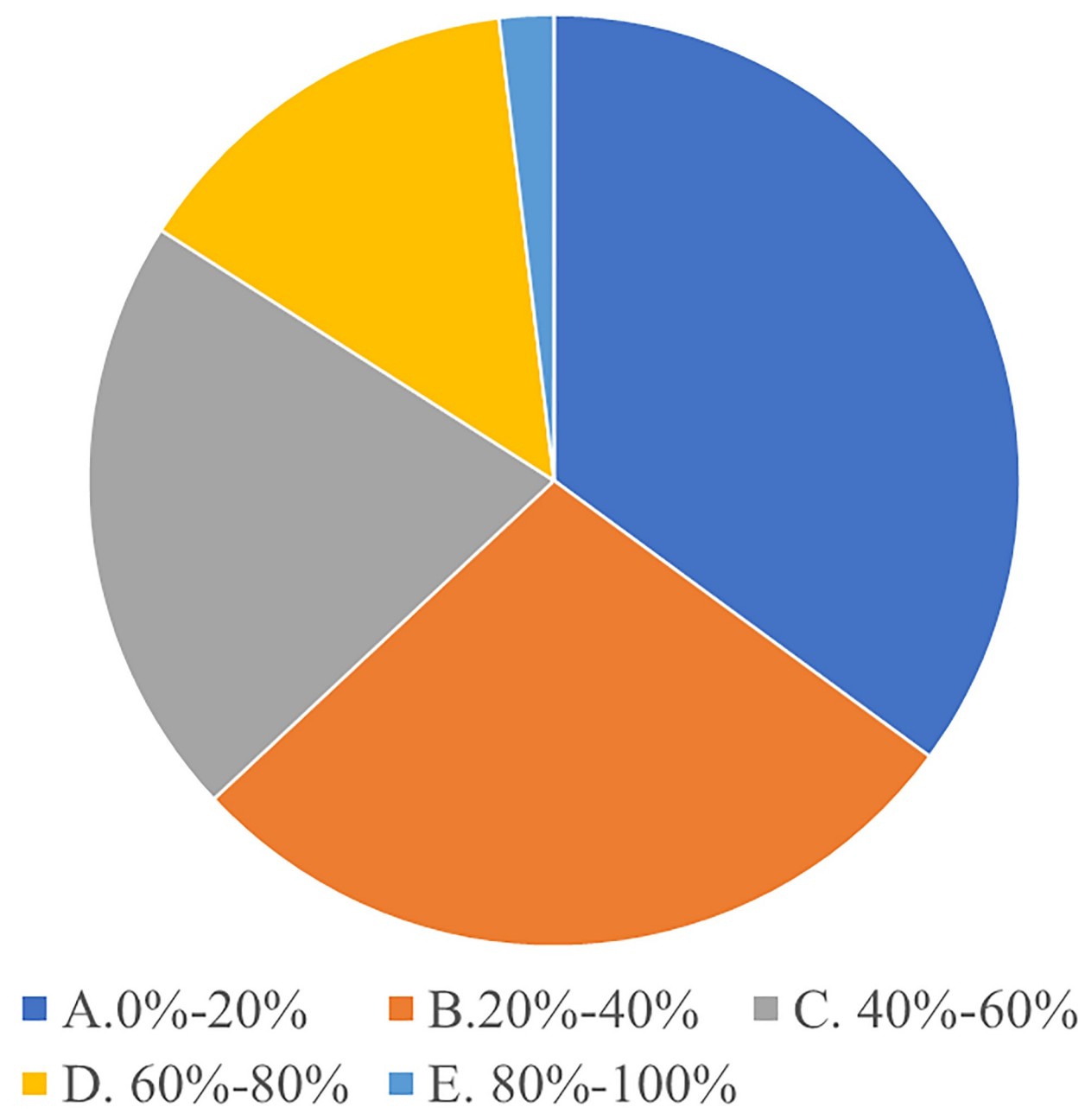

**Fig 2. Budget allocation for early childhood music education.**

escalating. According to market competition theories, the competition in this field is becoming increasingly fierce, with a multitude of music education institutions and a variety of courses emerging. To succeed in the competition, early childhood music education institutions need to continuously innovate their courses and teaching methods, such as introducing novel elements like gamified teaching and scenario simulation, providing personalized services that meet individual needs, to satisfy parents' and children's expectations to the greatest extent [57]. At the same time, they also need to actively respond to and anticipate market changes, adjusting strategies to maintain a competitive advantage.

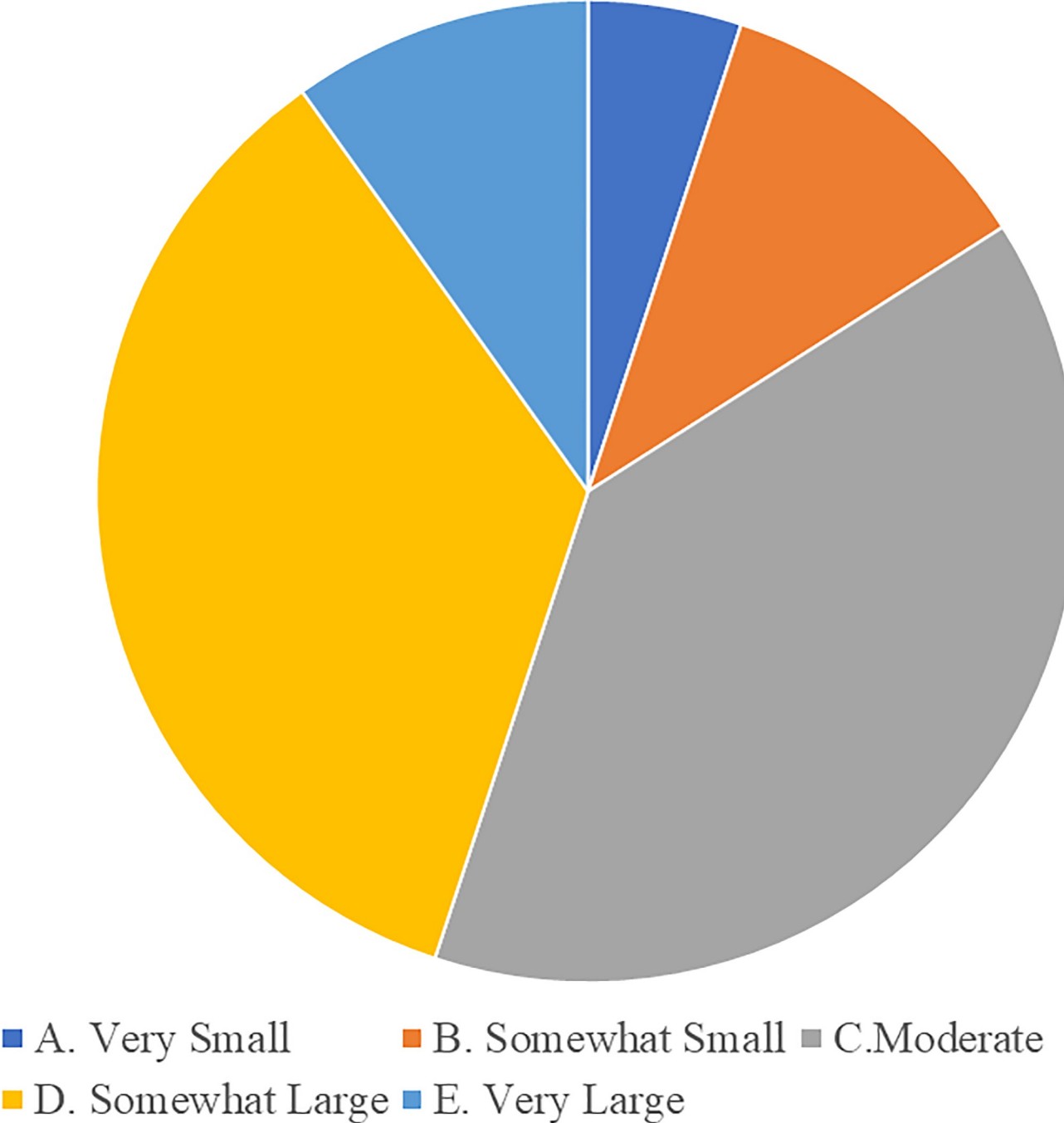

**Fig 3. Perceived impact of music education on child's development.**

**W2. Teacher shortage.** The excellent music teacher resources are limited and difficult to meet the rapid development needs of the industry. According to education industry theories, excellent music teachers are a key factor in the quality of early childhood music education [58]. However, the resources of excellent music teachers are limited, and market demand far exceeds supply. Faced with the problem of a shortage of teachers, early childhood music

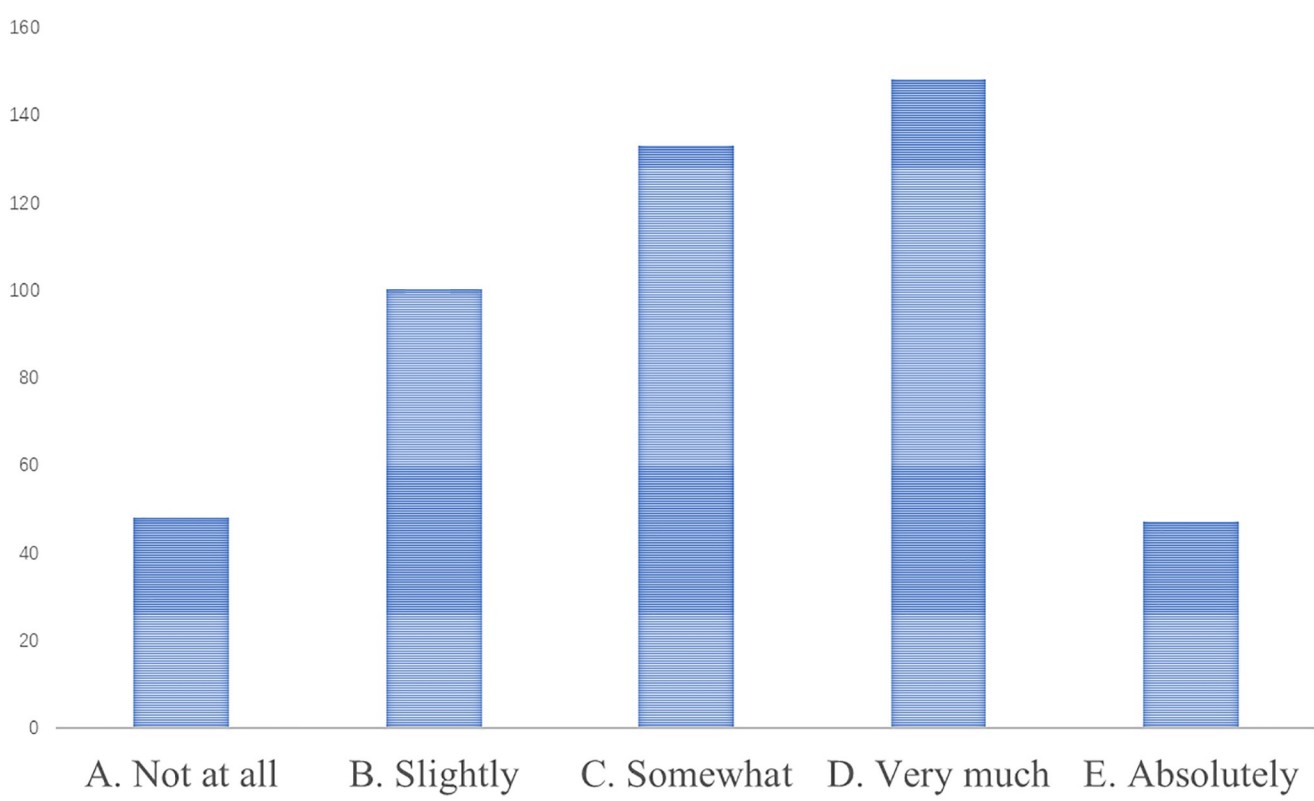

**Fig 4. Perception of the increase in early childhood music education institutions.**

education institutions need to continuously enhance teachers' professional level and teaching abilities through various channels such as training, recruitment, and cooperation [59]. Some music education institutions will establish cooperative relationships with music colleges, regularly inviting professional music teachers to conduct training, thus improving teachers' professional qualities and abilities, and solving the problem of teacher shortages.

**W3. Brand influence.** Compared to other mature education industries, early childhood music education has a lower brand awareness. According to brand building theories, brand recognition and influence are crucial to a company's competitiveness. However, the early childhood music education industry is relatively new and less well-known, making it difficult to stand out in the market. To enhance brand influence, early childhood music education institutions need to actively carry out marketing and promotional activities, strengthen the shaping of brand image, and establish good word-of-mouth and reputation. According to the results of the questionnaire survey (Figs 6 and 7), many music education institutions have expanded their brand influence and attracted more potential customer attention and trust by participating in various music competitions and public performances (supported by case studies).

**W4. Regulatory restrictions.** Policy and regulation may impose restrictions on the operations and admissions of early childhood music education institutions. For instance, at the end of 2022, the Ministry of Education of China and thirteen other departments issued the "Opinions on Regulating Non-Academic Off-Campus Training for Primary and Secondary School Students," indicating that the education industry needs to be supervised and managed by the government, with standards set and bottom-line requirements clarified. Early childhood music education institutions must strictly comply with related regulations. At the same time,

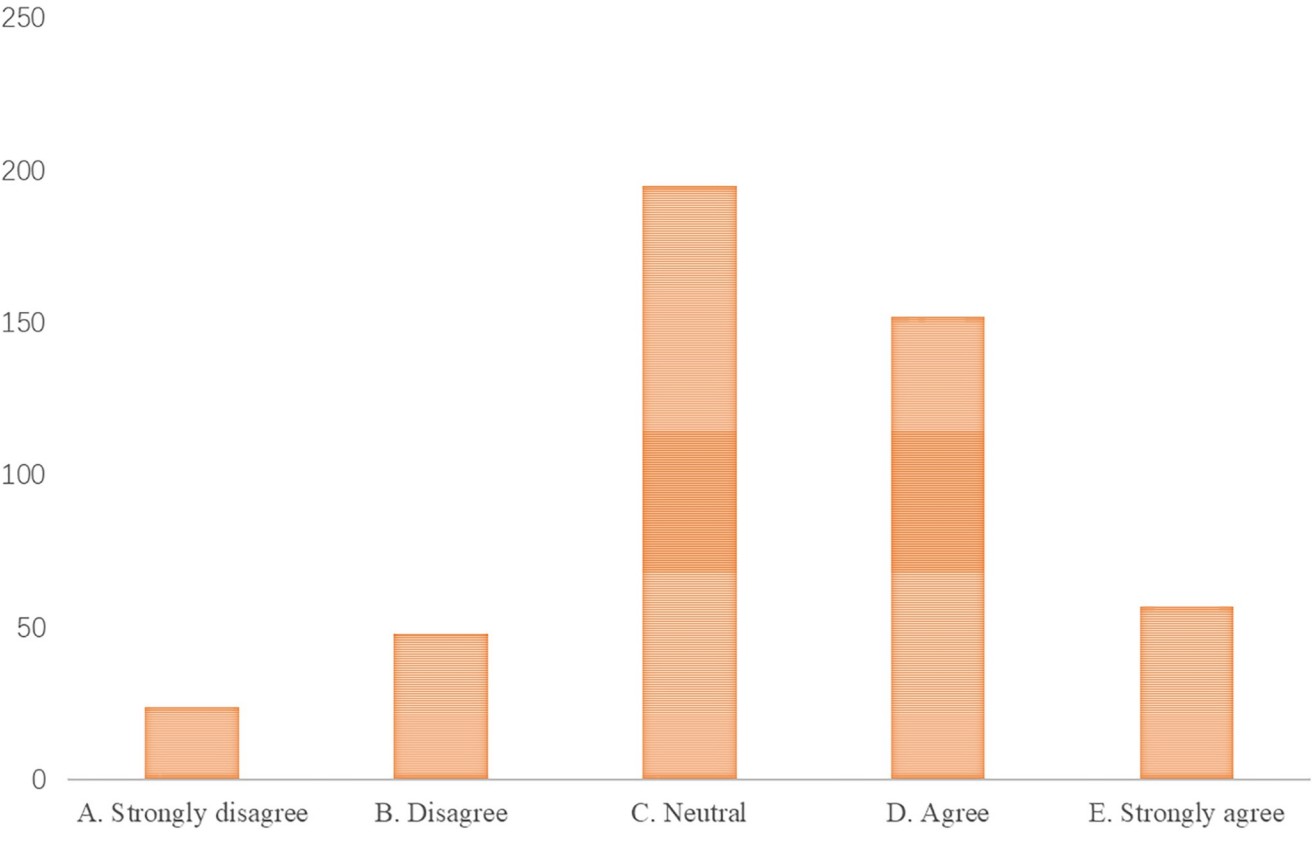

**Fig 5. Belief in the growth prospects of early childhood music education market.**

the uncertainty of policy and regulations may also impact the operations of the institutions. To deal with regulatory restrictions, early childhood music education institutions need to understand the latest policies timely, operate in compliance, and ensure that courses and teaching meet regulatory requirements.

### 4.3 Opportunities

**O1. Policy support.** The government's emphasis on early childhood education may bring favorable policies, providing support for the industry's development (See the policy support section in Appendix B: Table 11, Appendix file in the S1 File). According to education policy theories, the degree to which the government values early childhood education will directly influence the formulation and implementation of related policies. As society's awareness of early childhood education continues to increase, the government may strengthen its support for early childhood music education, such as giving priority in educational funding, teacher training, and the distribution of educational resources. In addition, the government's policy support can also help improve the overall image of the industry, attracting more parents to choose early childhood music education.

**O2. Technological innovation.** The application of online education, VR/AR, and other technologies is expected to bring new development opportunities to early childhood music education. According to the theory of educational technology, the continuous innovation of modern technology has brought new teaching methods and educational tools to early

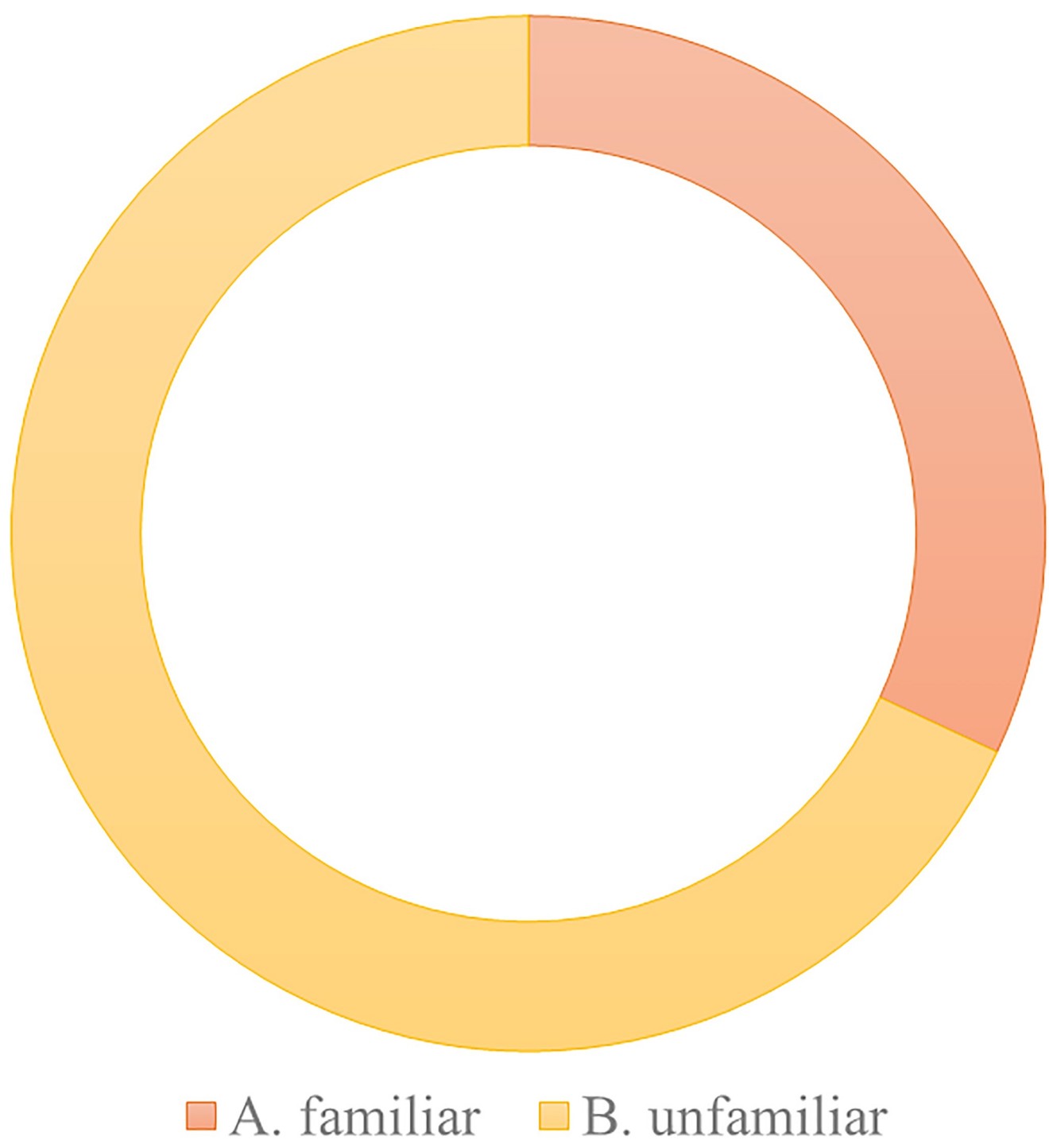

**Fig 6. Respondents' familiarity with well-known early childhood music education institutions.**

childhood music education [60]. The rise of online education platforms has allowed music education to break through the constraints of time and space, achieving remote teaching and resource sharing [61]. Meanwhile, Virtual Reality (VR) and Augmented Reality (AR) technologies bring a more vivid, interesting, and interactive experience to music teaching [62].

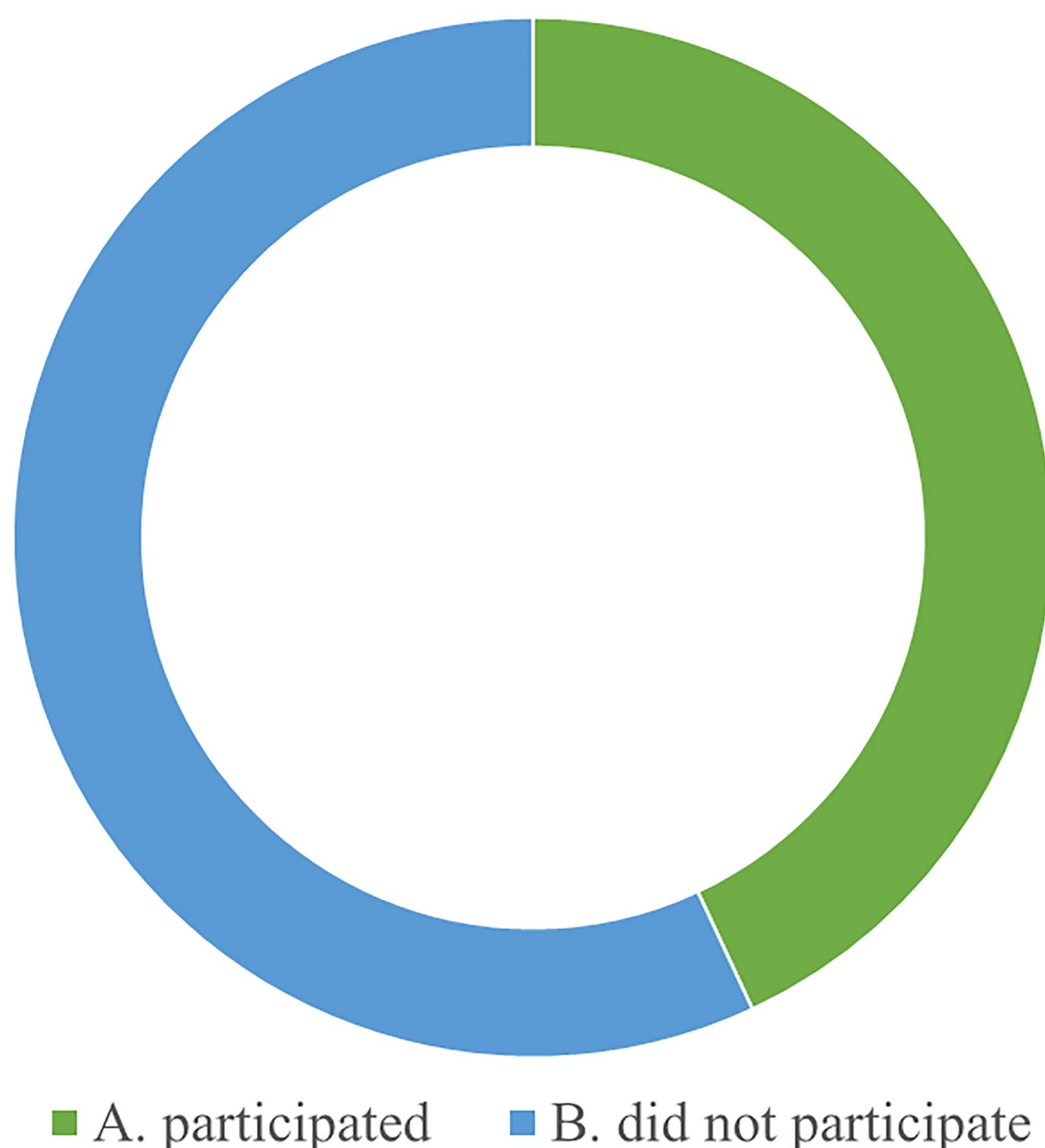

**Fig 7. Participation of respondents' children in music education activities.**

**O3. Market demand.** With parents' growing focus on comprehensive education for their children, the demand for music education is expected to continue to increase. According to market research theory, parents' investment in early childhood education is constantly increasing, and the need for comprehensive development of children is becoming increasingly

significant. As an important way to cultivate comprehensive skills, music education is gradually being recognized and valued by more and more parents. Therefore, the demand for early childhood music education is expected to continue to grow, driving the development of the industry (see Figs 1 and 2).

**O4. Cross-border cooperation.** The music education industry can carry out cross-border cooperation with other education sectors and cultural industries to expand market space. According to the theory of cross-border cooperation, collaboration across different fields can share resources, broaden market channels, and achieve mutual benefits. Early childhood music education, as a part of the cultural education sector, can cooperate with other art education, physical education and other fields to develop diversified courses and projects [56].

## 4.4 Threats

**T1. Economic fluctuations.** Changes in the economic environment may affect family education investment, and thus affect the demand for music education market [63]. According to economic theory, economic fluctuations can impact family income and consumption levels. In times of economic downturn, families may face financial pressures, and education expenditures may be cut, leading to a decrease in demand for music education [64]. However, during periods of economic recovery, family education investments may increase, and the demand for music education market may rebound [65].

**T2. Alternative products.** Other types of early childhood education programs, such as dance and art, may serve as substitutes for music education, competing for potential customers [66]. According to market competition theory, the early childhood education market is diversified, and parents may consider various options when selecting education programs. Some parents may prefer their children to participate in other types of arts education, such as dance courses or art training, instead of music education. To address the competition from substitutes, music education institutions need to highlight their uniqueness, providing curriculum content and teaching features that are different from other programs.

**T3. Regulatory risk.** Changes in policies and regulations could potentially have a negative impact on early childhood music education, such as increased supervision, tax policies, etc. (See the policy restrictions section in Appendix B: Table 11, Appendix file in the S1 File). According to legal theory, changes in government policies could pose new regulatory risks for early childhood music education institutions. For instance, the government might set higher standards for the registration of early education institutions, conduct stricter checks on teacher qualifications, and impose tighter control over tuition and charging behaviors. These policy changes may increase the operational costs and pressures for music education institutions. To mitigate regulatory risks, music education institutions need to stay updated with the latest policies, operate in compliance with regulations, and actively cooperate with government regulatory inspections and evaluations.

**T4. Industry standards.** The lack of unified teaching standards and evaluation systems within the industry could potentially lead to market chaos and a lack of confidence among parents [67]. According to theories on educational industry development, industry standards and evaluation systems are crucial for enhancing the development and norms of the entire industry. However, the standards and evaluation systems in the early childhood music education industry currently lack uniformity and authority, leading to a variety of courses and teaching contents on the market. Parents may feel hesitant and untrusting due to the lack of clear standards and references. To establish industry standards, music education institutions can participate in the standard-setting work of industry associations, contributing to the overall development of the industry.

**T5. Intellectual property.** Intellectual Property: Weak protection of intellectual property rights for music works and teaching materials could lead to serious piracy and copyright infringement, affecting the industry's reputation [68]. According to theories on intellectual property protection, safeguarding the intellectual property rights of music works and teaching materials is a vital means to protect the rights of creators and also key to maintaining the industry's reputation. If intellectual property rights for music works and teaching materials are inadequately protected, it could lead to rampant piracy, affecting the income of creators and damaging the reputation of the industry. To enhance the protection of intellectual property rights, music education institutions can actively cooperate with relevant agencies to strengthen copyright registration and protection, and at the same time advocate for students and teachers to respect intellectual property rights, enhancing awareness of intellectual property protection [69].

## Section 5: Numerical analysis process

In this section, we will delve into how to utilize the IFS-AHP-SWOT analysis method based on dynamic social networks to deeply research the development strategy of the early childhood music education industry. Detailed data used in this section are available from the authors upon request. First, based on the steps outlined in Section 3.4, we guided seven experts to construct the intuitive fuzzy preference relations for the 13 SWOT factors required for strategic analysis, as detailed in Section 3.2, using the intuitionistic fuzzy set theory introduced in Section 3.1 (refer to Table 2). Additionally, we developed a social network connection matrix among the experts as per the social network theory presented in Sections 3.3.2 and 3.3.3 (see Tables 4 and 5). Due to space constraints, we only detail the decision-making process of Expert No.1 in this article. For the decision-making process of other experts, interested readers can request from the authors.

Next, we proceed with Steps 3 and 4 to check whether the initial evaluation table of each expert satisfies the consistency of the intuitionistic fuzzy preference relationship (satisfying the $\varepsilon \leq 0.1$ condition). Then, using Algorithm 1 and Algorithm 2, we resolve inconsistent intuitionistic fuzzy preference relationships, or return the inconsistent relationships to the experts for re-evaluation until the evaluation results become acceptable. The revised evaluation from Expert No.1 is shown in Table 6:

Afterwards, we follow the guidance of Step 6 to calculate the attribute weights of the intuitionistic fuzzy preference relationships provided by each expert. Based on the evaluation table from Expert No.1, the attribute weights we derived are shown in Table 7:

Further, following Steps 6, we calculate the updated expert weights, as shown in Table 8:

Finally, according to Step 7, the comprehensive scores of SWOT that we obtained are shown in Table 9:

We use the total strength of advantages (S), the total strength of weaknesses (W), the total strength of opportunities (O), and the total strength of threats (T) as four variables to construct a four-dimensional coordinate system. In this coordinate system, based on the comprehensive scores obtained from Table 8, we determine the corresponding points of S, W, O, and T respectively, and then connect these four points in sequence to get our strategic quadrilateral (as shown in Fig 7). This strategic quadrilateral reflects the comprehensive effect of the four factors of strengths, weaknesses, opportunities, and threats, and it is an important basis for enterprises to make strategic choices. At the same time, the centroid of the quadrilateral reflects the result of the comprehensive effect of the four factors (as indicated by the red dot in Fig 8).

**Table 4. Initial evaluation table of expert No.1.**

| | | | | | |
|---|---|---|---|---|---|
| SWOT intuition fuzzy preference relation | | | | | |
| | S | W | O | T | |
| S | (0.5,0.5) | (0.12,0.87) | (0.09,0.87) | (0.08,0.77) | |
| W | (0.87,0.12) | (0.5,0.5) | (0.06,0.7) | (0.07,0.85) | |
| O | (0.87,0.09) | (0.7,0.06) | (0.5,0.5) | (0.11,0.83) | |
| T | (0.77,0.08) | (0.85,0.07) | (0.83,0.11) | (0.5,0.5) | |
| Strengths intuition fuzzy preference relation | | | | | |
| | S1 | S2 | S3 | S4 | |
| S1 | (0.5,0.5) | (0.41,0.43) | (0.31,0.02) | (0.29,0.2) | |
| S2 | (0.43,0.41) | (0.5,0.5) | (0.09,0.16) | (0.32,0.41) | |
| S3 | (0.02,0.31) | (0.16,0.09) | (0.5,0.5,0) | (0.39,0.43) | |
| S4 | (0.2,0.29) | (0.41,0.32) | (0.43,0.39) | (0.5,0.5) | |
| Weaknesses intuition fuzzy preference relation | | | | | |
| | W1 | W2 | W3 | W4 | |
| W1 | (0.5,0.5) | (0.71,0.19) | (0.72,0.04) | (0.66,0.04) | |
| W2 | (0.19,0.71) | (0.5,0.5) | (0.01,0.99) | (0.22,0.23) | |
| W3 | (0.04,0.72) | (0.99,0.01) | (0.5,0.5) | (0.04,0.5) | |
| W4 | (0.04,0.66) | (0.23,0.22) | (0.5,0.04) | (0.5,0.5) | |
| Opportunities intuition fuzzy preference relation | | | | | |
| | O1 | O2 | O3 | O4 | |
| O1 | (0.5,0.5) | (0.21,0.67) | (0.89,0.04) | (0.37,0.56) | |
| O2 | (0.67,0.21) | (0.5,0.5) | (0.3,0.7) | (0.25,0.14) | |
| O3 | (0.04,0.89) | (0.7,0.3) | (0.5,0.5) | (0.4,0.59) | |
| O4 | (0.56,0.37) | (0.14,0.25) | (0.59,0.4) | (0.5,0.5) | |
| Threats intuition fuzzy preference relation | | | | | |
| | T1 | T2 | T3 | T4 | T5 |
| T1 | (0.5,0.5) | (0.08,0.64) | (0.22,0.61) | (0.3,0.5) | (0.04,0.87) |
| T2 | (0.64,0.08) | (0.5,0.5) | (0.06,0.93) | (0.58,0.04) | (0.15,0.68) |
| T3 | (0.61,0.22) | (0.93,0.06) | (0.5,0.5) | (0.45,0.45) | (0.26,0.39) |
| T4 | (0.5,0.3) | (0.04,0.58) | (0.45,0.45) | (0.5,0.5) | (0.2,0.41) |
| T5 | (0.87,0.04) | (0.68,0.15) | (0.39,0.26) | (0.41,0.2) | (0.5,0.5) |

**Table 5. Initial social network.**

| | e1 | e2 | e3 | e4 | e5 | e6 | e7 |
|---|---|---|---|---|---|---|---|
| e1 | | (0.07, 0.34) | (0.16, 0.55) | (0.35, 0.4) | (0.08, 0.13) | (0.33, 0.56) | (0.6, 0.32) |
| e2 | (0.34, 0.07) | | (0.74, 0.04) | (0.23, 0.76) | (0.45, 0.33) | (0.14, 0.54) | (0.01, 0.99) |
| e3 | (0.55, 0.16) | (0.04, 0.74) | | (0.13, 0.62) | (0.07, 0.7) | (0.1, 0.63) | (0.07, 0.58) |
| e4 | (0.4, 0.35) | (0.76, 0.23) | (0.62, 0.13) | | (0.29, 0.1) | (0.29, 0.66) | (0.25, 0.33) |
| e5 | (0.13, 0.08) | (0.33, 0.45) | (0.7, 0.07) | (0.1, 0.29) | | (0.2, 0.65) | (0.32, 0.31) |
| e6 | (0.56, 0.33) | (0.54, 0.14) | (0.63, 0.1) | (0.66, 0.29) | (0.65, 0.2) | | (0.09, 0.62) |
| e7 | (0.32, 0.6) | (0.99, 0.01) | (0.58, 0.07) | (0.33, 0.25) | (0.31, 0.32) | (0.62, 0.09) | |

**Table 6. Evaluation table of Expert No.1 meeting consistency.**

| | S | W | O | T |
|---|---|---|---|---|
| | | | SWOT intuition fuzzy preference relation | |
| S | (0.5,0.5) | (0.12,0.87) | (0.02,0.92) | (0.0,0.94) |
| W | (0.87,0.12) | (0.5,0.5) | (0.06,0.7) | (0.02,0.9) |
| O | (0.92,0.02) | (0.7,0.06) | (0.5,0.5) | (0.11,0.83) |
| T | (0.94,0.0) | (0.9,0.02) | (0.83,0.11) | (0.5,0.5) |

| | S1 | S2 | S3 | S4 |
|---|---|---|---|---|
| | | | Strengths intuition fuzzy preference relation | |
| S1 | (0.5,0.5) | (0.41,0.43) | (0.08,0.1) | (0.1,0.1) |
| S2 | (0.43,0.41) | (0.5,0.5) | (0.09,0.16) | (0.08,0.15) |
| S3 | (0.1,0.08) | (0.16,0.09) | (0.5,0.5) | (0.39,0.43) |
| S4 | (0.1,0.1) | (0.15,0.08) | (0.43,0.39) | (0.5,0.5) |

| | W1 | W2 | W3 | W4 |
|---|---|---|---|---|
| | | | Weaknesses intuition fuzzy preference relation | |
| W1 | (0.5,0.5) | (0.71,0.19) | (0.03,0.94) | (0.0,0.84) |
| W2 | (0.19,0.71) | (0.5,0.5) | (0.01,0.99) | (0.0,0.98) |
| W3 | (0.94,0.03) | (0.99,0.01) | (0.5,0.5) | (0.04,0.5) |
| W4 | (0.84,0.0) | (0.98,0.0) | (0.5,0.04) | (0.5,0.5) |

| | O1 | O2 | O3 | O4 |
|---|---|---|---|---|
| | | | Opportunities intuition fuzzy preference relation | |
| O1 | (0.5,0.5) | (0.21,0.67) | (0.12,0.8) | (0.11,0.76) |
| O2 | (0.67,0.21) | (0.5,0.5) | (0.3,0.7) | (0.23,0.75) |
| O3 | (0.8,0.12) | (0.7,0.3) | (0.5,0.5) | (0.4,0.59) |
| O4 | (0.76,0.11) | (0.75,0.23) | (0.59,0.4) | (0.5,0.5) |

| | T1 | T2 | T3 | T4 | T5 |
|---|---|---|---|---|---|
| | | | Threats intuition fuzzy preference relation | | |
| T1 | (0.5,0.5) | (0.08,0.64) | (0.02,0.95) | (0.02,0.85) | (0.0,0.76) |
| T2 | (0.64,0.08) | (0.5,0.5) | (0.06,0.93) | (0.06,0.88) | (0.03,0.78) |
| T3 | (0.95,0.02) | (0.93,0.06) | (0.5,0.5) | (0.45,0.45) | (0.18,0.37) |
| T4 | (0.85,0.02) | (0.88,0.06) | (0.45,0.45) | (0.5,0.5) | (0.2,0.41) |
| T5 | (0.76,0.0) | (0.78,0.03) | (0.37,0.18) | (0.41,0.2) | (0.5,0.5) |

**Table 7. Attribute weights derived from the evaluation table of Expert No.1.**

| S | W | O | T | |
|---|---|---|---|---|
| (0.08, 0.9) | (0.17, 0.76) | (0.26, 0.65) | (0.37, 0.55) | |
| S1 | S2 | S3 | S4 | |
| (0.09, 0.37) | (0.1, 0.38) | (0.1, 0.36) | (0.1, 0.35) | |
| W1 | W2 | W3 | W4 | |
| (0.14, 0.79) | (0.08, 0.89) | (0.28, 0.59) | (0.32, 0.52) | |
| O1 | O2 | O3 | O4 | |
| (0.11, 0.83) | (0.2, 0.76) | (0.29, 0.67) | (0.31, 0.64) | |
| T1 | T2 | T3 | T4 | T5 |
| (0.04, 0.88) | (0.09, 0.83) | (0.21, 0.66) | (0.2, 0.66) | (0.2, 0.61) |

**Table 8. Expert weights.**

| e1 | e2 | e3 | e4 | e5 | e6 | e7 |
|---|---|---|---|---|---|---|
| 0.1394 | 0.1494 | 0.1394 | 0.1561 | 0.1327 | 0.1444 | 0.1386 |

**Table 9. Comprehensive evaluation scores.**

| S | W | O | T |
|---|---|---|---|
| 1.5076 | 0.9168 | 0.946 | 0.7837 |

As can be seen from Fig 8, the centroid is located in the first quadrant, belonging to the exploratory strategic area. For institutions focusing on the early childhood music education industry, this suggests adopting an Opportunity-Strength (OS) based development strategy. Therefore, in order to leverage the strengths of early childhood music education institutions and seize potential market opportunities, institutions need to formulate a comprehensive set of strategies. This includes, first and foremost, the creation of a comprehensive and specific mental health education plan. The content of the plan can cover the design and promotion of

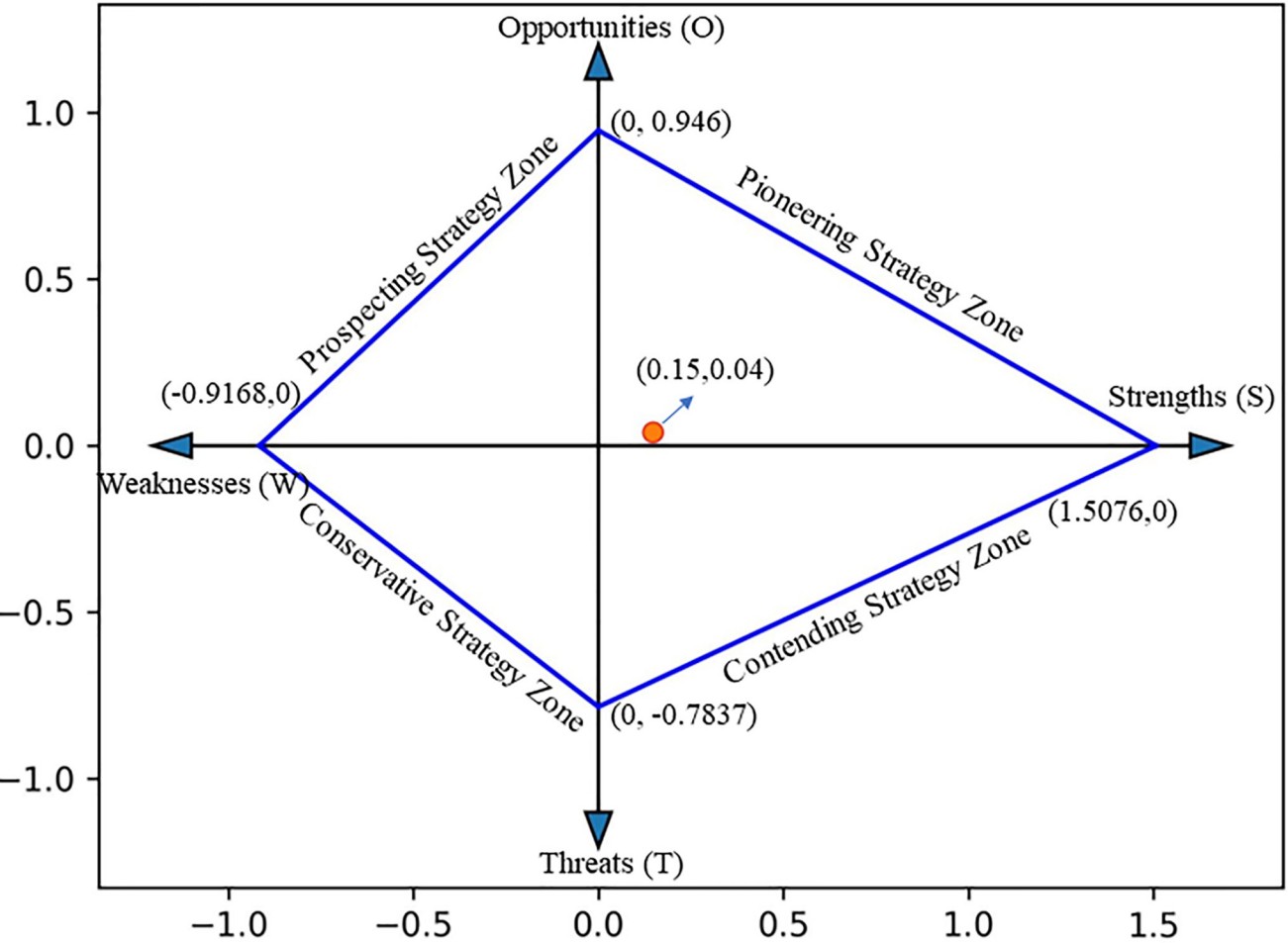

**Fig 8. Development strategy quadrilateral for early childhood music education industry.**

specific mental health courses, providing professional psychological counseling services, and promoting the importance of mental health through various methods such as seminars and lectures. During this process, policy support, as a valuable resource, should be fully utilized to promote the importance and development of mental health education. At the same time, we need to pay attention to and introduce new teaching technologies and methods. For example, by using cutting-edge virtual reality (VR) and augmented reality (AR) technologies, an immersive learning experience can be created for students. Or, by using artificial intelligence (AI) technology, personalized learning tools and resources can be developed to adapt to the different needs of each student, improving the effectiveness and quality of education. Additionally, institutions should actively seek to establish cooperative relationships with enterprises or institutions in other fields. This includes choosing partners with complementary advantages, establishing clear cooperation agreements, and jointly developing new courses or products to promote educational innovation and diversification. Lastly, institutions should actively seek government funding and support. This might involve active communication and cooperation with government departments, such as applying for relevant government funding programs and securing policy benefits for educational development.

During the implementation of these strategies, we must always pay attention to their effects and make timely adjustments and optimizations according to the actual situation to ensure the effectiveness and rationality of the strategies.

## Section 6: Conclusions

Early childhood music education holds a place in today's education system due to its significant impact on children's psychological, social skills, and cognitive development. However, its current state and importance are recognized differently across various regions and cultures worldwide. In many developed countries and regions, early childhood music education has been widely accepted and integrated into the early education system. Teachers adopt an edutainment approach, allowing children to learn music through singing, dancing, and playing instruments. This teaching method has been proven to enhance children's language skills, memory, coordination, creativity, and social skills.

In recent years, with the change in educational concepts and social-economic development, more and more people are beginning to realize the importance of music education and are starting to invest in this field. Additionally, with the advancement of technology, the internet and digital education tools have opened up new possibilities for early childhood music education. Online music education platforms, mobile applications, and virtual reality technology have all started to be used in early childhood music education, creating new avenues for the popularization and optimization of music education.

The persistent and high-quality development of the early childhood music education industry cannot be separated from precise and comprehensive strategic analysis. Our research uses an analytical technique improved through social networks, which incorporates the opinions of education experts, industry practitioners, and policy decision-makers. Through the integrated application of Intuitionistic Fuzzy Sets (IFS), Analytic Hierarchy Process (AHP), and Strengths, Weaknesses, Opportunities, and Threats (SWOT) analysis, it provides deep insights for strategic planning in the field of early childhood music education (In order to further illustrate the robustness and effectiveness of the method used in this article, we have added the corresponding comparative analysis in Appendix C, Appendix file in the S1 File). The research results reveal that relevant institutions in early childhood music education should seize their strengths and opportunities (OS) and adopt corresponding development strategies. According to the current data, the market size of China's music education industry continues to exhibit a

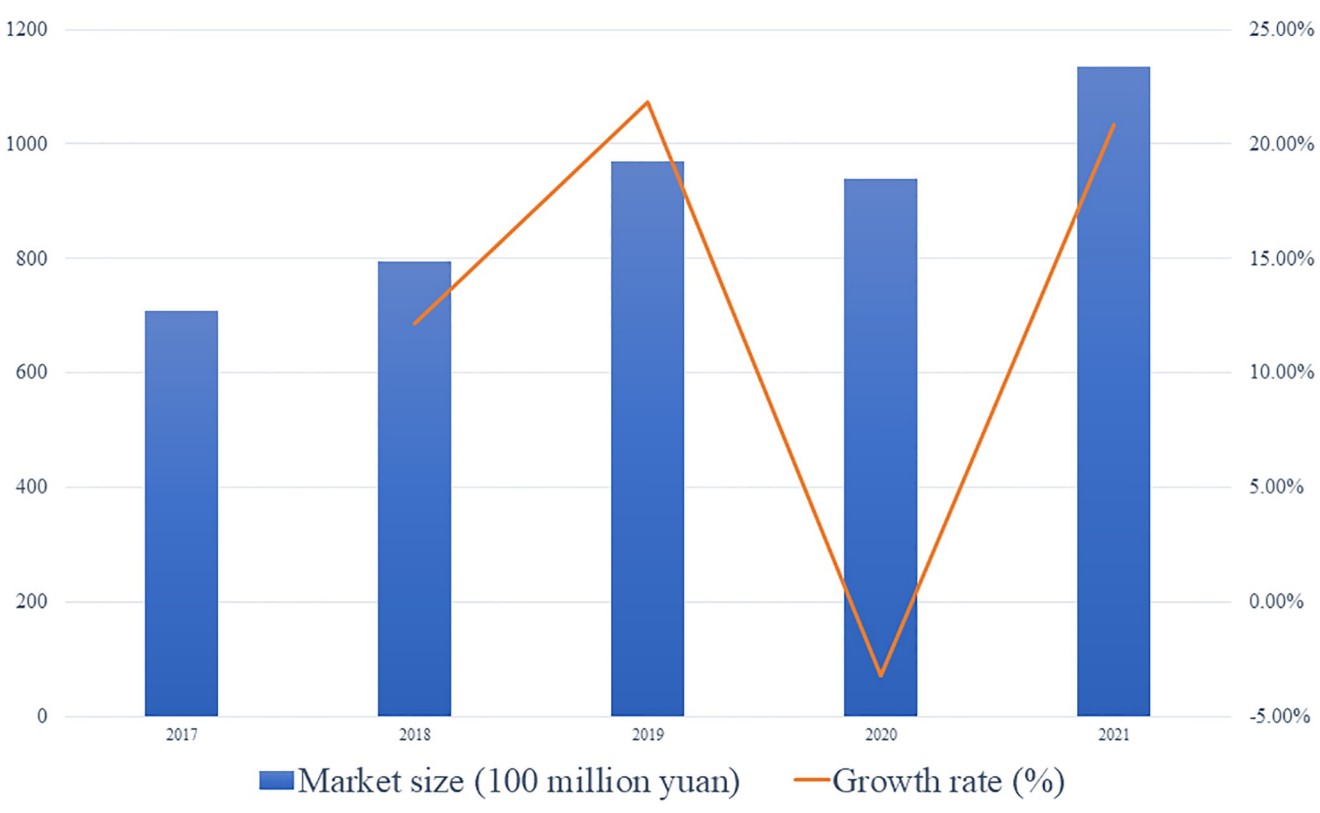

**Fig 9. Changes in market size of China's music education industry from 2017 to 2021.**

growth trend. As of 2021, the projected market size for China's music education is expected to reach 113.38 billion yuan, representing a year-on-year increase of 20.84% (the data is sourced from the "2022–2027 China Music Education Industry Operation Situation and Future Development Trend Forecast Report" released by the Huajing Industrial Research Institute). The Chinese music education industry is closely intertwined with demographics, policies, and economic development. In the future, with the gradual improvement of music education policies and the gradual economic recovery post-pandemic, it is anticipated that the market size of China's music education industry will continue to grow steadily, aligning with the conclusions presented in this article (Fig 9).

To better implement this Strengths and Opportunities-based development strategy, we recommend that early childhood music education institutions should first strengthen their core strengths, which may include unique teaching methods, experienced educators, or a good community reputation. In addition, they should look for and seize opportunities for development, such as taking advantage of technological advancements to develop online teaching, develop music education applications, or establish partnerships with other educational institutions to share resources. Our research also emphasizes the importance of continuous evaluation and adjustment of strategies. Early childhood music education institutions should regularly use the analysis method we proposed to understand changes in their own strengths and weaknesses, identify new opportunities and challenges, and then adjust their development strategies accordingly. At the same time, policymakers also have a responsibility to create a favorable environment for the development of early childhood music education. They can achieve this goal by providing financial support, optimizing education policies, or raising public awareness of the importance of early childhood music education.

In conclusion, we believe that by implementing a development strategy based on strengths and opportunities, the early childhood music education industry will be able to achieve sustainable high-quality development, provide better music education services for children, and contribute to the development of society.

## Supporting information

**S1 File.**
(ZIP)

## Author Contributions

**Conceptualization:** Yuanyang Yue.

**Data curation:** Yuanyang Yue.

**Formal analysis:** Yuanyang Yue.

**Investigation:** Yuanyang Yue.

**Methodology:** Yuanyang Yue.

**Project administration:** Yuanyang Yue.

**Software:** Xiaoyan Shen.

**Supervision:** Xiaoyan Shen.

**Validation:** Xiaoyan Shen.

**Visualization:** Xiaoyan Shen.

**Writing – original draft:** Xiaoyan Shen.

**Writing – review & editing:** Xiaoyan Shen.

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
