## [Decision Letter · Decision Letter 0]

8 Sep 2023

PONE-D-23-24110Development Strategy of Early Childhood Music Education Industry: An IFS-AHP-SWOT Analysis Based on Dynamic Social NetworkPLOS ONE

Dear Dr. Shen,

Thank you for submitting your manuscript to PLOS ONE. After careful consideration, we feel that it has merit but does not fully meet PLOS ONE’s publication criteria as it currently stands. Therefore, we invite you to submit a revised version of the manuscript that addresses the points raised during the review process.

We look forward to receiving your revised manuscript.

Kind regards,

Muhammet Gul, Ph.D.

Academic Editor

PLOS ONE

Journal Requirements:

2. For studies reporting research involving human participants, PLOS ONE requires authors to confirm that this specific study was reviewed and approved by an institutional review board (ethics committee) before the study began. Please provide the specific name of the ethics committee/IRB that approved your study, or explain why you did not seek approval in this case.

7. We note you have included a table to which you do not refer in the text of your manuscript. Please ensure that you refer to Table 9 in your text; if accepted, production will need this reference to link the reader to the Table.

Reviewers' comments:

Reviewer's Responses to Questions

**Comments to the Author**

1. Is the manuscript technically sound, and do the data support the conclusions?

Reviewer #1: Yes

Reviewer #2: Partly

2. Has the statistical analysis been performed appropriately and rigorously? 

Reviewer #1: Yes

Reviewer #2: N/A

3. Have the authors made all data underlying the findings in their manuscript fully available?

Reviewer #1: No

Reviewer #2: Yes

4. Is the manuscript presented in an intelligible fashion and written in standard English?

Reviewer #1: Yes

Reviewer #2: Yes

5. Review Comments to the Author

Reviewer #1: Dear Editor,

I have reviewed the manuscript entitled “Development Strategy of Early Childhood Music Education Industry: An IFS-AHP-SWOT Analysis Based on Dynamic Social Network”, which proposed an analytical method based on dynamic social networks in conjunction with Intuitionistic Fuzzy Sets (IFS), Analytic Hierarchy Process (AHP), and Strengths, Weaknesses, Opportunities, and Threats (SWOT) to navigate the challenges and explore growth strategies for the early childhood music education industry. Although the idea is of interesting, the structure and scientific discussions of the current version of the manuscript are to be improved for publication. Anyway, there are some comments which may be helpful for authors:

1) There are some spelling and grammatical errors, which dictate an in-depth review.

2) The research has not a comprehensive structure or framework, and seems a discrete application of various independent methods.

3) It is better to more describe the problem, and the relation between IFS, AHP, and SWOT methods.

4) It is suggested to present a flow chart or a pseudo code for better understanding the analyzing process.

5) Some ports including data and information need a reliable reference.

6) How could the authors compare their finding with real situation?

7) It is better to compare the findings of this research with those of previous studies.

Kind regards,

S. Mahdevari

Reviewer #2: This study aims to determine the Development Strategy of the Early Childhood Music Education Industry. The study proposes an analytical method based on dynamic social networks together with Intuitive Fuzzy Sets (IFS), Analytical Hierarchy Process (AHP) and Strengths, Weaknesses, Opportunities and Threats (SWOT) analysis. It is stated how the study can contribute to industry practitioners, policy makers and researchers.

I propose to consider the issues I have presented in the following items.

* In the literature review, explanatory information such as the original aspects of the studies and the methods used can also be presented in the form of a table. In addition, the limitations of existing studies should be emphasized. Explain why IF-AHP-SWOT integration is needed.

*The organization of the study should be reviewed. After the theoretical parts are presented, numerical results should be included under the application title. The steps in the theoretical part should correspond to the steps of the application parts. Step-by-step results should be presented. In addition, how IF-AHP and SWOT integration is done, a flow chart should be created.

*Chinese expressions should be translated into English in the fourth table.

*To demonstrate the robustness of the proposed methodology, a comperative study and sensitivity analysis should be presented.

6. PLOS authors have the option to publish the peer review history of their article (what does this mean?). If published, this will include your full peer review and any attached files.

Reviewer #1: No

Reviewer #2: No

---

## [Author Response · Author response to Decision Letter 0]

19 Oct 2023

Manuscript Number: PONE-D-23-24110

Title: Development Strategy of Early Childhood Music Education Industry: An IFS-AHP-SWOT Analysis Based on Dynamic Social Network

Submitted for publication in PLOS ONE by Yuanyang Yue and Xiaoyan Shen*

We would like to extend our sincere gratitude to you for carefully reviewing our paper, including the helpful comments and constructive suggestions you put forward to improve the paper. We have revised the paper based on your helpful suggestions and comments, which are written in italics below. Our responses (blue font) and revisions (black font with bold emphasis) are provided below for each of your comments.

Reviewers' comments:

Reviewer's Responses to Questions

Comments to the Author

1. Is the manuscript technically sound, and do the data support the conclusions?

Reviewer #1: Yes

Reviewer #2: Partly

Response: Thank you very much for your comment. We rechecked the data and calculation procedures, and described the data acquisition channels in the supporting material. At the same time, it is also stated in the article that relevant materials can also be requested directly from the author.

2. Has the statistical analysis been performed appropriately and rigorously?

Reviewer #1: Yes

Reviewer #2: N/A

Response: Thank you very much for your comment. The statistical analyzes we conduct in the paper are appropriate and rigorous.

3. Have the authors made all data underlying the findings in their manuscript fully available?

Reviewer #1: No

Reviewer #2: Yes

Response: Thank you very much for your comment. We rechecked the data and calculation procedures, and described the data acquisition channels in the supporting material. At the same time, it is also stated in the article that relevant materials can also be requested directly from the author. 

4. Is the manuscript presented in an intelligible fashion and written in standard English?

Reviewer #1: Yes

Reviewer #2: Yes

Response: Thank you very much for your comment. We have re-examined the language expression in this article to ensure that the language is clear, correct and unambiguous.

5. Review Comments to the Author

Reviewer #1: Dear Editor,

I have reviewed the manuscript entitled “Development Strategy of Early Childhood Music Education Industry: An IFS-AHP-SWOT Analysis Based on Dynamic Social Network”, which proposed an analytical method based on dynamic social networks in conjunction with Intuitionistic Fuzzy Sets (IFS), Analytic Hierarchy Process (AHP), and Strengths, Weaknesses, Opportunities, and Threats (SWOT) to navigate the challenges and explore growth strategies for the early childhood music education industry. Although the idea is of interesting, the structure and scientific discussions of the current version of the manuscript are to be improved for publication. Anyway, there are some comments which may be helpful for authors:

1. There are some spelling and grammatical errors, which dictate an in-depth review.

Response: Thank you so much for your careful check. We have made further corrections to address language issues in the text such as spelling and grammar. As follows:

Original sentence: Therefore, in order to promote the all-round development of young children, it is imperative to carry out music education.

After modification: Therefore, to promote the all-round development of young children, it is imperative to carry out music education.

Original sentence: However, in teaching practice, there is often too much emphasis on skill training, neglecting the cultivation of children's interest in music and aesthetic ability. When conducting music activities, we usually only educate children to sing and play games, neglecting the exploration of other aspects of music

After modification: Nevertheless, in teaching practice, there is often an excessive emphasis on skill training, disregarding the nurturing of children's interest in music and their aesthetic abilities. During music activities, the focus typically remains on teaching children to sing and play games, overlooking the exploration of other facets of music

Original sentence: This study comprehensively applies dynamic social networks, IFS, AHP, and SWOT analysis methods to provide a systematic analytical framework and guiding recommendations for the development strategy of the early childhood music education industry.

After modification: This study comprehensively applies dynamic social networks, IFS, AHP, and SWOT analysis methods to offer a systematic analytical framework and guiding recommendations for the development strategy of the early childhood music education industry.

2. The research has not a comprehensive structure or framework, and seems a discrete application of various independent methods.

Response: Thank you for your rigorous comment. SWOT analysis is a common method for business and industry strategic analysis. However, because it cannot determine the degree of influence of each factor, it is often used in conjunction with the analytic hierarchy process (AHP-SWOT method). We introduce the intuitive fuzzy set theory into the AHP-SWOT method to make up for the limitations caused by limited expert knowledge and subjective evaluation criteria. And we have also improved the method of aggregating multiple expert judgment information, taking into account the social network relationships formed by different trust relationships, and solving potential abnormal results due to individual differences. Through the comprehensive application of multiple methods, we can more accurately point out the direction for the sustainable development of the early childhood music education industry. We explain it in more detail in the paper and provide a corresponding comprehensive framework. The specific modifications are as follows: 

1. Introduction

[…]

With the rapid development of the early childhood music education industry, conducting effective strategic analysis to promote its sustainable high-quality growth has become particularly important. SWOT analysis is a commonly used method for business and industry strategic analysis [12,13]. However, because it cannot determine the degree of influence of each factor, it is often used in combination with AHP (AHP-SWOT method) [14,15,16]. Although AHP has proven to be effective and simple in dealing with multi-criteria decision-making problems, it cannot fully address inherent uncertainties and fuzziness. Therefore, this paper introduces Intuitionistic Fuzzy Set Theory into the AHP-SWOT method (IFS-AHP-SWOT method) to compensate for the limitations caused by experts' limited knowledge and subjective evaluation criteria [17,18]. Strategic analysis for the early childhood music education industry requires the collaborative judgment of multiple experts. Therefore, this paper improves the method of aggregating the judgment information of multiple experts while considering social network relationships formed by different trust relationships, addressing the potential abnormal results due to individual differences [19,20]. This improved IFS-AHP-SWOT analysis method based on dynamic social networks, combined with the strategic analysis viewpoints of educators, practitioners, and policymakers, provides strong support for the sustainable development of the early childhood music education industry. The research results indicate that institutions in the early childhood music education industry should adopt development strategies based on strengths and opportunities (SO). This study comprehensively applies dynamic social networks, IFS, AHP, and SWOT analysis methods to offer a systematic analytical framework and guiding recommendations for the development strategy of the early childhood music education industry. It is hoped that this research will provide valuable reference for early childhood music education practitioners, policymakers, and researchers, promoting the continuous development and progress of the industry.

3.4 Decision-making steps

Step 1: Define the evaluation criteria system for SWOT analysis of the early childhood music education industry's development strategy. Then proceed to the next step.

Step 2: Have each strategic analysis expert use the IFS (Formula 1) to compare each criterion pairwise, creating multiple intuitionistic fuzzy preference relation matrices. Continue to the next step.

Step 3: Use Formula 12 to check the consistency of the experts' provided intuitionistic fuzzy preference relations. If all intuitionistic fuzzy preference relations are consistent and acceptable, move on to Step 5; otherwise, proceed to Step 4.

Step 4: Use Algorithm 1 and Algorithm 2 to resolve inconsistent intuitionistic fuzzy preference relations (or return the inconsistent relations to experts for reevaluation until they become acceptable). Then, proceed to the next step.

Step 5: Based on all the intuitionistic fuzzy preference relations, calculate the attribute weights for each criterion in the evaluation criteria system. Calculate expert weights based on social networks (Formulas 16-20) and move on to the next step.

Step 6: Utilize expert weights and attribute weights to aggregate all intuitionistic fuzzy preference relation matrices (Formulas 21-23) to obtain the group intuitionistic fuzzy preference relations. Check for group consensus; if achieved, continue to the next step. Otherwise, return to Step 2, where experts with significant differences in group intuitionistic fuzzy preference relations will reevaluate, while the remaining experts will reassess their social network relationships.

Step 7: Apply Formulas 3 to 7 to aggregate the group intuitionistic fuzzy preference relations and derive the final scores for S (Strengths), W (Weaknesses), O (Opportunities), and T (Threats).

The above decision-making steps can be summarized as shown in Figure 1.

Figure 1. Decision-making steps flow chart

3. It is better to more describe the problem, and the relation between IFS, AHP, and SWOT methods.

Response: Thank you for your rigorous comment. We have explained the relationship between IFS, AHP and SWOT methods in detail in the introduction and literature review. The specific modifications are as follows: 

1. Introduction

[…]

With the rapid development of the early childhood music education industry, conducting effective strategic analysis to promote its sustainable high-quality growth has become particularly important. SWOT analysis is a commonly used method for business and industry strategic analysis [12,13]. However, because it cannot determine the degree of influence of each factor, it is often used in combination with AHP (AHP-SWOT method) [14,15,16]. Although AHP has proven to be effective and simple in dealing with multi-criteria decision-making problems, it cannot fully address inherent uncertainties and fuzziness. Therefore, this paper introduces Intuitionistic Fuzzy Set Theory into the AHP-SWOT method (IFS-AHP-SWOT method) to compensate for the limitations caused by experts' limited knowledge and subjective evaluation criteria [17,18]. Strategic analysis for the early childhood music education industry requires the collaborative judgment of multiple experts. Therefore, this paper improves the method of aggregating the judgment information of multiple experts while considering social network relationships formed by different trust relationships, addressing the potential abnormal results due to individual differences [19,20]. This improved IFS-AHP-SWOT analysis method based on dynamic social networks, combined with the strategic analysis viewpoints of educators, practitioners, and policymakers, provides strong support for the sustainable development of the early childhood music education industry. The research results indicate that institutions in the early childhood music education industry should adopt development strategies based on strengths and opportunities (SO). This study comprehensively applies dynamic social networks, IFS, AHP, and SWOT analysis methods to offer a systematic analytical framework and guiding recommendations for the development strategy of the early childhood music education industry. It is hoped that this research will provide valuable reference for early childhood music education practitioners, policymakers, and researchers, promoting the continuous development and progress of the industry.

2.Literature Review

The above research can be summarized as shown in Table 1, as follows:

Table 1. Summary of literature on research methods

 SWOT AHP Fuzzy decision-making Social network

[36,37,38,39,40] ✓ ✓

[41,42] ✓ ✓ 

[43,44] ✓ ✓ ✓ 

This paper ✓ ✓ ✓ ✓

In summary, current research on early childhood music often focuses on the promotion of various aspects of music education for young children, such as social and perceptual benefits. However, it frequently overlooks the broader, long-term development strategy of the early childhood music education industry within the context of the current dynamic social environment. This gap in research leaves a critical need for a comprehensive approach to address the industry's future growth and adaptability.

One of the shortcomings in the current research landscape is the prevalent use of intuitive and somewhat disjointed methods, such as intuitionistic fuzzy AHP (Analytic Hierarchy Process) and SWOT (Strengths, Weaknesses, Opportunities, and Threats) analysis. These methods are often employed independently, failing to harness the full potential of their complementary attributes. Moreover, they generally disregard the significant impact of social networks and interactions among expert reviewers in the process of shaping industry development strategies.

To address these limitations, this paper proposes a novel approach that combines dynamic social networks, IFS (Intuitionistic Fuzzy Set), AHP, and SWOT analysis methods. This integrated methodology aims to establish a comprehensive and cohesive framework for analyzing the development strategy of the early childhood music education industry. By merging these diverse techniques, this research endeavors to create a more holistic perspective on industry growth and provide a more effective and adaptable strategy. It recognizes the importance of considering both the intrinsic factors identified through SWOT analysis and the dynamic, interrelated social factors that shape the industry's trajectory, as assessed through social network analysis. In doing so, it offers valuable guidance for stakeholders and decision-makers within the early childhood music education sector to make informed, data-driven decisions that can lead to sustained success in a rapidly evolving environment.

4. It is suggested to present a flow chart or a pseudo code for better understanding the analyzing process.

Response: Thank you for your significant reminding. We have rearranged the decision-making process steps in the text and added a flowchart to better understand the analysis process. The specific modifications are as follows: 

3.4 Decision-making steps

Step 1: Define the evaluation criteria system for SWOT analysis of the early childhood music education industry's development strategy. Then proceed to the next step.

Step 2: Have each strategic analysis expert use the IFS (Formula 1) to compare each criterion pairwise, creating multiple intuitionistic fuzzy preference relation matrices. Continue to the next step.

Step 3: Use Formula 12 to check the consistency of the experts' provided intuitionistic fuzzy preference relations. If all intuitionistic fuzzy preference relations are consistent and acceptable, move on to Step 5; otherwise, proceed to Step 4.

Step 4: Use Algorithm 1 and Algorithm 2 to resolve inconsistent intuitionistic fuzzy preference relations (or return the inconsistent relations to experts for reevaluation until they become acceptable). Then, proceed to the next step.

Step 5: Based on all the intuitionistic fuzzy preference relations, calculate the attribute weights for each criterion in the evaluation criteria system. Calculate expert weights based on social networks (Formulas 16-20) and move on to the next step.

Step 6: Utilize expert weights and attribute weights to aggregate all intuitionistic fuzzy preference relation matrices (Formulas 21-23) to obtain the group intuitionistic fuzzy preference relations. Check for group consensus; if achieved, continue to the next step. Otherwise, return to Step 2, where experts with significant differences in group intuitionistic fuzzy preference relations will reevaluate, while the remaining experts will reassess their social network relationships.

Step 7: Apply Formulas 3 to 7 to aggregate the group intuitionistic fuzzy preference relations and derive the final scores for S (Strengths), W (Weaknesses), O (Opportunities), and T (Threats).

The above decision-making steps can be summarized as shown in Figure 1.

Figure 1. Decision-making steps flow chart

5. Some ports including data and information need a reliable reference.

Response: Thank you for your significant reminding. We emphasize again in the article that relevant data can be obtained from the author. And we have also submitted relevant data documents, which can be viewed in the supporting information.

5. Numerical analysis process

In this section, we will delve into how to utilize the IFS-AHP-SWOT analysis method based on dynamic social networks to deeply research the development strategy of the early childhood music education industry. Detailed data used in this section are available from the authors upon request. First, following Steps 1 and 2 mentioned in Section 3.4, we guide seven experts (refer to Table 2) to determine the intuitionistic fuzzy preference relations for the 13 SWOT factors required for strategic analysis (refer to Table 4), and construct a social network connection matrix between them (refer to Table 5). Due to space constraints, we only detail the decision-making process of Expert No.1 in this article. For the decision-making process of other experts, interested readers can request from the authors.

Supporting Information.

Data can be found in the supporting information

6. How could the authors compare their finding with real situation?

Response: We gratefully thanks for the precious time the reviewer spent making constructive remarks. The conclusions we draw indicate that institutions related to early childhood music education should seize their own advantages and opportunities and adopt corresponding development strategies. Judging from the data in recent years, the market size of China's music education industry has indeed shown a continuous growth trend. We explain it in more detail in the article, and the specific modifications are as follows: 

6. Conclusions

[…].

The persistent and high-quality development of the early childhood music education industry cannot be separated from precise and comprehensive strategic analysis. Our research uses an analytical technique improved through social networks, which incorporates the opinions of education experts, industry practitioners, and policy decision-makers. Through the integrated application of Intuitionistic Fuzzy Sets (IFS), Analytic Hierarchy Process (AHP), and Strengths, Weaknesses, Opportunities, and Threats (SWOT) analysis, it provides deep insights for strategic planning in the field of early childhood music education (In order to further illustrate the robustness and effectiveness of the method used in this article, we have added the corresponding comparative analysis in Appendix C). The research results reveal that relevant institutions in early childhood music education should seize their strengths and opportunities (OS) and adopt corresponding development strategies. According to the current data, the market size of China's music education industry continues to exhibit a growth trend. As of 2021, the projected market size for China's music education is expected to reach 113.38 billion yuan, representing a year-on-year increase of 20.84% (the data is sourced from the "2022-2027 China Music Education Industry Operation Situation and Future Development Trend Forecast Report" released by the Huajing Industrial Research Institute). The Chinese music education industry is closely intertwined with demographics, policies, and economic development. In the future, with the gradual improvement of music education policies and the gradual economic recovery post-pandemic, it is anticipated that the market size of China's music education industry will continue to grow steadily, aligning with the conclusions presented in this article.

Figure 9. Changes in market size of China’s music education industry from 2017 to 2021

To better implement this Strengths and Opportunities-based development strategy, we recommend that early childhood music education institutions should first strengthen their core strengths, which may include unique teaching methods, experienced educators, or a good community reputation.

7. It is better to compare the findings of this research with those of previous studies.

Response: Thank you for your rigorous consideration. We used different methods to compare this study with previous research. This not only illustrates the validity of the conclusions of this article, but also illustrates the robustness of the method in this article. The specific modifications are as follows:

Appendix C: Comparative analysis

We validate the robustness and effectiveness of the methods used in this paper from two perspectives. One is to revalidate the cases in this paper using the methods from relevant literature, and the other is to validate the cases from the relevant literature using the methods presented in this paper. However, different evaluation values are used in different papers. For instance, Dağdeviren and Yüksel (2011) used linguistic fuzzy numbers, Papapostolou et al. (2020) used triangular fuzzy numbers, and Tavana et al. (2016) used triangular intuitionistic fuzzy numbers. Therefore, it is necessary to unify these evaluation values into a common fuzzy number evaluation using the following methods:

1. Conversion of Different Fuzzy Number Evaluation Values

Let the intuitionistic fuzzy number be represented as a=(μ_a,ν_a ); the triangular intuitionistic fuzzy number be represented as b=〈(▁b,b,¯b);δ_b,ε_b 〉, where δ_b represents the maximum membership degree, and ε_b represents the minimum non-membership degree. The conversion relationship between the intuitionistic fuzzy number a and the triangular intuitionistic fuzzy number b is shown in formulas 24-25: 

μ_a={█(((x-▁b) δ_b)⁄((b-▁b) ) if ▁b≤x<b@δ_b if x=b@((¯b-x) δ_b)⁄((¯b-b) ) ifb<x≤¯b@0 if x<▁b or x>¯b)┤ (24)

ν_a={█(((b-x+(x-▁b) δ_b ))⁄((b-▁b) ) if ▁b≤x<b@ε_b if x=b@((x-b+(¯b-x) δ_b ))⁄((¯b-b) ) ifb<x≤¯b@1 if x<▁b or x>¯b)┤ (25)

The conversion relationships between intuitionistic fuzzy numbers, linguistic fuzzy numbers, and triangular fuzzy numbers are shown in Table 11 (Wu and Xu, 2015; Eftekhary et al., 2012):

Table 11. Conversion Table for Indicator Evaluation Linguistic Term Sets

Linguistic fuzzy terms Triangular fuzzy numbers Intuitive fuzzy numbers

Very Good/Very High (9,10,10) (0.85,0.1)

Good/High (7,9,10) (0.75,0.15)

Fairly Good/Fairly High (5,7,9) (0.65,0.25)

Moderate/Medium (3,5,7) (0.5,0.4)

Fairly Poor/Fairly Low (1,3,5) (0.35,0.55)

Poor/Low (0,1,3) (0.25,0.65)

Very Poor/Very Low (0,0,1) (0.15,0.8)

2. Use the methods in related papers to re-verify the case in this Paper

 S W O T

This Paper 1.5076 0.9168 0.946 0.7837

Dağdeviren and Yüksel (2011) 1.2538 1.1404 0.7231 0.7324

Papapostolou et al., 2020 1.2647 1.3844 1.212 0.9464

Tavana et al., 2016 1.1421 1.005 0.848 0.7197

3. Use the method in this Paper to verify the cases in related papers

The original result of Dağdeviren and Yüksel (2011) Work system 1 has a score of 0.557 and Work system 2 has a score of 0.372. Work system 1 is better than Work system 2.

The result of Dağdeviren and Yüksel (2011) verified by the method of this Paper Work system 1 has a score of 0.742 and Work system 2 has a score of 0.531. Work system 1 is better than Work system 2.

The original result of Papapostolou et al., 2020 SO strategy ≻ ST strategy ≻ WO strategy ≻ WT strategy

The result of Papapostolou et al., 2020 verified by the method of this Paper SO strategy ≻ WO strategy ≻ ST strategy ≻ WT strategy

The original result of Tavana et al., 2016 The strengths more important than weaknesses, opportunities, and threats. 

The result of Tavana et al., 2016 verified by the method of this Paper Strengths and opportunities are almost equally important, but both are far more important than weaknesses and threats.

From the above results, it can be observed that whether revalidating the cases in this paper using methods from relevant literature or validating the cases from relevant literature using the methods presented in this paper, the final results, although with slight variations, do not exhibit significant changes. This indicates the robustness and effectiveness of the methods employed in this paper across different cases and evaluation values.

Wu, Z., & Xu, J. (2015). Possibility distribution-based approach for MAGDM with hesitant fuzzy linguistic information. IEEE transactions on cybernetics, 46(3), 694-705.

Eftekhary, M., Safari, S., Shojaee, M., Assarian, M., & Karimi, I. (2012). Identifying customers needs on electronic services of bank using fuzzy QFD approach. Aust. J. Basic Appl. Sci, 6, 287-296.

Dağdeviren, M., & Yüksel, İ. (2008). Developing a fuzzy analytic hierarchy process (AHP) model for behavior-based safety management. Information sciences, 178(6), 1717-1733.

Papapostolou, A., Karakosta, C., Apostolidis, G., & Doukas, H. (2020). An AHP-SWOT-Fuzzy TOPSIS approach for achieving a cross-border RES cooperation. Sustainability, 12(7), 2886.

Tavana, M., Zareinejad, M., Di Caprio, D., & Kaviani, M. A. (2016). An integrated intuitionistic fuzzy AHP and SWOT method for outsourcing reverse logistics. Applied soft computing, 40, 544-557.

Reviewer #2: This study aims to determine the Development Strategy of the Early Childhood Music Education Industry. The study proposes an analytical method based on dynamic social networks together with Intuitive Fuzzy Sets (IFS), Analytical Hierarchy Process (AHP) and Strengths, Weaknesses, Opportunities and Threats (SWOT) analysis. It is stated how the study can contribute to industry practitioners, policy makers and researchers.

I propose to consider the issues I have presented in the following items.

1. In the literature review, explanatory information such as the original aspects of the studies and the methods used can also be presented in the form of a table. In addition, the limitations of existing studies should be emphasized. Explain why IF-AHP-SWOT integration is needed.

Response: Thank you for your rigorous consideration. We have added relevant discussion of why IF-AHP-SWOT integration is needed in both the introduction and the literature review. And in the review we also added relevant comparison tables and added the limitations of existing studies. The specific modifications are as follows:

1. Introduction

[…]

With the rapid development of the early childhood music education industry, conducting effective strategic analysis to promote its sustainable high-quality growth has become particularly important. SWOT analysis is a commonly used method for business and industry strategic analysis [12,13]. However, because it cannot determine the degree of influence of each factor, it is often used in combination with AHP (AHP-SWOT method) [14,15,16]. Although AHP has proven to be effective and simple in dealing with multi-criteria decision-making problems, it cannot fully address inherent uncertainties and fuzziness. Therefore, this paper introduces Intuitionistic Fuzzy Set Theory into the AHP-SWOT method (IFS-AHP-SWOT method) to compensate for the limitations caused by experts' limited knowledge and subjective evaluation criteria [17,18]. Strategic analysis for the early childhood music education industry requires the collaborative judgment of multiple experts. Therefore, this paper improves the method of aggregating the judgment information of multiple experts while considering social network relationships formed by different trust relationships, addressing the potential abnormal results due to individual differences [19,20]. This improved IFS-AHP-SWOT analysis method based on dynamic social networks, combined with the strategic analysis viewpoints of educators, practitioners, and policymakers, provides strong support for the sustainable development of the early childhood music education industry. The research results indicate that institutions in the early childhood music education industry should adopt development strategies based on strengths and opportunities (SO). This study comprehensively applies dynamic social networks, IFS, AHP, and SWOT analysis methods to offer a systematic analytical framework and guiding recommendations for the development strategy of the early childhood music education industry. It is hoped that this research will provide valuable reference for early childhood music education practitioners, policymakers, and researchers, promoting the continuous development and progress of the industry.

2.Literature Review

The above research can be summarized as shown in Table 1, as follows:

Table 1. Summary of literature on research methods

 SWOT AHP Fuzzy decision-making Social network

[36,37,38,39,40] ✓ ✓

[41,42] ✓ ✓ 

[43,44] ✓ ✓ ✓ 

This paper ✓ ✓ ✓ ✓

In summary, current research on early childhood music often focuses on the promotion of various aspects of music education for young children, such as social and perceptual benefits. However, it frequently overlooks the broader, long-term development strategy of the early childhood music education industry within the context of the current dynamic social environment. This gap in research leaves a critical need for a comprehensive approach to address the industry's future growth and adaptability.

One of the shortcomings in the current research landscape is the prevalent use of intuitive and somewhat disjointed methods, such as intuitionistic fuzzy AHP (Analytic Hierarchy Process) and SWOT (Strengths, Weaknesses, Opportunities, and Threats) analysis. These methods are often employed independently, failing to harness the full potential of their complementary attributes. Moreover, they generally disregard the significant impact of social networks and interactions among expert reviewers in the process of shaping industry development strategies.

To address these limitations, this paper proposes a novel approach that combines dynamic social networks, IFS (Intuitionistic Fuzzy Set), AHP, and SWOT analysis methods. This integrated methodology aims to establish a comprehensive and cohesive framework for analyzing the development strategy of the early childhood music education industry. By merging these diverse techniques, this research endeavors to create a more holistic perspective on industry growth and provide a more effective and adaptable strategy. It recognizes the importance of considering both the intrinsic factors identified through SWOT analysis and the dynamic, interrelated social factors that shape the industry's trajectory, as assessed through social network analysis. In doing so, it offers valuable guidance for stakeholders and decision-makers within the early childhood music education sector to make informed, data-driven decisions that can lead to sustained success in a rapidly evolving environment.

2.The organization of the study should be reviewed. After the theoretical parts are presented, numerical results should be included under the application title. The steps in the theoretical part should correspond to the steps of the application parts. Step-by-step results should be presented. In addition, how IF-AHP and SWOT integration is done, a flow chart should be created.

Response: Thank you for your rigorous consideration. We divide the theoretical part of the paper into three aspects, one is intuitive fuzzy, the other is intuitive fuzzy preference relationship, and the other is weight calculation. We indicate which formulas are used in the decision-making steps in Section 3.4 and add the corresponding flow chart. In the application chapter of Chapter 5, we strictly follow the decision-making steps and indicate the methods and theories used. The specific modifications are as follows: 

5. Numerical analysis process

In this section, we will delve into how to utilize the IFS-AHP-SWOT analysis method based on dynamic social networks to deeply research the development strategy of the early childhood music education industry. Detailed data used in this section are available from the authors upon request. First, based on the steps outlined in Section 3.4, we guided seven experts to construct the intuitive fuzzy preference relations for the 13 SWOT factors required for strategic analysis, as detailed in Section 3.2, using the intuitionistic fuzzy set theory introduced in Section 3.1 (refer to Table 2). Additionally, we developed a social network connection matrix among the experts as per the social network theory presented in Sections 3.3.2 and 3.3.3 (see Table 5). Due to space constraints, we only detail the decision-making process of Expert No.1 in this article. For the decision-making process of other experts, interested readers can request from the authors

3.4 Decision-making steps

Step 1: Define the evaluation criteria system for SWOT analysis of the early childhood music education industry's development strategy. Then proceed to the next step.

Step 2: Have each strategic analysis expert use the IFS (Formula 1) to compare each criterion pairwise, creating multiple intuitionistic fuzzy preference relation matrices. Continue to the next step.

Step 3: Use Formula 12 to check the consistency of the experts' provided intuitionistic fuzzy preference relations. If all intuitionistic fuzzy preference relations are consistent and acceptable, move on to Step 5; otherwise, proceed to Step 4.

Step 4: Use Algorithm 1 and Algorithm 2 to resolve inconsistent intuitionistic fuzzy preference relations (or return the inconsistent relations to experts for reevaluation until they become acceptable). Then, proceed to the next step.

Step 5: Based on all the intuitionistic fuzzy preference relations, calculate the attribute weights for each criterion in the evaluation criteria system. Calculate expert weights based on social networks (Formulas 16-20) and move on to the next step.

Step 6: Utilize expert weights and attribute weights to aggregate all intuitionistic fuzzy preference relation matrices (Formulas 21-23) to obtain the group intuitionistic fuzzy preference relations. Check for group consensus; if achieved, continue to the next step. Otherwise, return to Step 2, where experts with significant differences in group intuitionistic fuzzy preference relations will reevaluate, while the remaining experts will reassess their social network relationships.

Step 7: Apply Formulas 3 to 7 to aggregate the group intuitionistic fuzzy preference relations and derive the final scores for S (Strengths), W (Weaknesses), O (Opportunities), and T (Threats).

The above decision-making steps can be summarized as shown in Figure 1.

Figure 1. Decision-making steps flow chart

3.Chinese expressions should be translated into English in the fourth table.

Response: Thank you so much for your careful check. We translated the Chinese expressions in the fourth table into English, and because new pictures were added, the serial number was changed to Figure 5. The modification is as follows:

Table 5 Initial social network

4.To demonstrate the robustness of the proposed methodology, a comperative study and sensitivity analysis should be presented.

Response: Thank you for your rigorous consideration. We used different methods to compare this study with previous research. This not only illustrates the validity of the conclusions of this article, but also illustrates the robustness of the method in this article. The specific modifications are as follows:

Appendix C: Comparative analysis

We validate the robustness and effectiveness of the methods used in this paper from two perspectives. One is to revalidate the cases in this paper using the methods from relevant literature, and the other is to validate the cases from the relevant literature using the methods presented in this paper. However, different evaluation values are used in different papers. For instance, Dağdeviren and Yüksel (2011) used linguistic fuzzy numbers, Papapostolou et al. (2020) used triangular fuzzy numbers, and Tavana et al. (2016) used triangular intuitionistic fuzzy numbers. Therefore, it is necessary to unify these evaluation values into a common fuzzy number evaluation using the following methods:

1. Conversion of Different Fuzzy Number Evaluation Values

Let the intuitionistic fuzzy number be represented as a=(μ_a,ν_a ); the triangular intuitionistic fuzzy number be represented as b=〈(▁b,b,¯b);δ_b,ε_b 〉, where δ_b represents the maximum membership degree, and ε_b represents the minimum non-membership degree. The conversion relationship between the intuitionistic fuzzy number a and the triangular intuitionistic fuzzy number b is shown in formulas 24-25: 

μ_a={█(((x-▁b) δ_b)⁄((b-▁b) ) if ▁b≤x<b@δ_b if x=b@((¯b-x) δ_b)⁄((¯b-b) ) ifb<x≤¯b@0 if x<▁b or x>¯b)┤ (24)

ν_a={█(((b-x+(x-▁b) δ_b ))⁄((b-▁b) ) if ▁b≤x<b@ε_b if x=b@((x-b+(¯b-x) δ_b ))⁄((¯b-b) ) ifb<x≤¯b@1 if x<▁b or x>¯b)┤ (25)

The conversion relationships between intuitionistic fuzzy numbers, linguistic fuzzy numbers, and triangular fuzzy numbers are shown in Table 11 (Wu and Xu, 2015; Eftekhary et al., 2012):

Table 11. Conversion Table for Indicator Evaluation Linguistic Term Sets

Linguistic fuzzy terms Triangular fuzzy numbers Intuitive fuzzy numbers

Very Good/Very High (9,10,10) (0.85,0.1)

Good/High (7,9,10) (0.75,0.15)

Fairly Good/Fairly High (5,7,9) (0.65,0.25)

Moderate/Medium (3,5,7) (0.5,0.4)

Fairly Poor/Fairly Low (1,3,5) (0.35,0.55)

Poor/Low (0,1,3) (0.25,0.65)

Very Poor/Very Low (0,0,1) (0.15,0.8)

2. Use the methods in related papers to re-verify the case in this Paper

 S W O T

This Paper 1.5076 0.9168 0.946 0.7837

Dağdeviren and Yüksel (2011) 1.2538 1.1404 0.7231 0.7324

Papapostolou et al., 2020 1.2647 1.3844 1.212 0.9464

Tavana et al., 2016 1.1421 1.005 0.848 0.7197

3. Use the method in this Paper to verify the cases in related papers

The original result of Dağdeviren and Yüksel (2011) Work system 1 has a score of 0.557 and Work system 2 has a score of 0.372. Work system 1 is better than Work system 2.

The result of Dağdeviren and Yüksel (2011) verified by the method of this Paper Work system 1 has a score of 0.742 and Work system 2 has a score of 0.531. Work system 1 is better than Work system 2.

The original result of Papapostolou et al., 2020 SO strategy ≻ ST strategy ≻ WO strategy ≻ WT strategy

The result of Papapostolou et al., 2020 verified by the method of this Paper SO strategy ≻ WO strategy ≻ ST strategy ≻ WT strategy

The original result of Tavana et al., 2016 The strengths more important than weaknesses, opportunities, and threats. 

The result of Tavana et al., 2016 verified by the method of this Paper Strengths and opportunities are almost equally important, but both are far more important than weaknesses and threats.

From the above results, it can be observed that whether revalidating the cases in this paper using methods from relevant literature or validating the cases from relevant literature using the methods presented in this paper, the final results, although with slight variations, do not exhibit significant changes. This indicates the robustness and effectiveness of the methods employed in this paper across different cases and evaluation values.

Wu, Z., & Xu, J. (2015). Possibility distribution-based approach for MAGDM with hesitant fuzzy linguistic information. IEEE transactions on cybernetics, 46(3), 694-705.

Eftekhary, M., Safari, S., Shojaee, M., Assarian, M., & Karimi, I. (2012). Identifying customers needs on electronic services of bank using fuzzy QFD approach. Aust. J. Basic Appl. Sci, 6, 287-296.

Dağdeviren, M., & Yüksel, İ. (2008). Developing a fuzzy analytic hierarchy process (AHP) model for behavior-based safety management. Information sciences, 178(6), 1717-1733.

Papapostolou, A., Karakosta, C., Apostolidis, G., & Doukas, H. (2020). An AHP-SWOT-Fuzzy TOPSIS approach for achieving a cross-border RES cooperation. Sustainability, 12(7), 2886.

Tavana, M., Zareinejad, M., Di Caprio, D., & Kaviani, M. A. (2016). An integrated intuitionistic fuzzy AHP and SWOT method for outsourcing reverse logistics. Applied soft computing, 40, 544-557.

---

## [Decision Letter · Decision Letter 1]

21 Nov 2023

Development Strategy of Early Childhood Music Education Industry: An IFS-AHP-SWOT Analysis Based on Dynamic Social Network

PONE-D-23-24110R1

Dear Dr. Shen,

We’re pleased to inform you that your manuscript has been judged scientifically suitable for publication and will be formally accepted for publication once it meets all outstanding technical requirements.

Kind regards,

Muhammet Gul, Ph.D.

Academic Editor

PLOS ONE

Additional Editor Comments (optional):

Reviewers' comments:

Reviewer's Responses to Questions

**Comments to the Author**

1. If the authors have adequately addressed your comments raised in a previous round of review and you feel that this manuscript is now acceptable for publication, you may indicate that here to bypass the “Comments to the Author” section, enter your conflict of interest statement in the “Confidential to Editor” section, and submit your "Accept" recommendation.

Reviewer #1: All comments have been addressed

2. Is the manuscript technically sound, and do the data support the conclusions?

Reviewer #1: Yes

3. Has the statistical analysis been performed appropriately and rigorously? 

Reviewer #1: Yes

4. Have the authors made all data underlying the findings in their manuscript fully available?

Reviewer #1: Yes

5. Is the manuscript presented in an intelligible fashion and written in standard English?

Reviewer #1: Yes

6. Review Comments to the Author

Reviewer #1: (No Response)

7. PLOS authors have the option to publish the peer review history of their article (what does this mean?). If published, this will include your full peer review and any attached files.

Reviewer #1: **Yes: **Satar Mahdevari

---

## [Editor Report · Acceptance letter]

24 Nov 2023

PONE-D-23-24110R1 

Development Strategy of Early Childhood Music Education Industry: An IFS-AHP-SWOT Analysis Based on Dynamic Social Network 

Dear Dr. Shen:

I'm pleased to inform you that your manuscript has been deemed suitable for publication in PLOS ONE. Congratulations! Your manuscript is now with our production department. 

Kind regards, 

on behalf of

Dr. Muhammet Gul 

Academic Editor

PLOS ONE